# Learning Efficient Planning-based Rewards for Imitation Learning

## Abstract

Imitation learning from limited demonstrations is challenging. Most inverse reinforcement learning (IRL) methods are unable to perform as good as the demonstrator, especially in a high-dimensional environment, e.g, the Atari domain. To address this challenge, we propose a novel reward learning method, which streamlines a differential planning module with dynamics modeling. Our method learns useful planning computations with a meaningful reward function that focuses on the resulting region of an agent executing an action. Such a planning-based reward function leads to policies with better generalization ability. Empirical results with multiple network architectures and reward instances show that our method can outperform state-of-the-art IRL methods on multiple Atari games and continuous control tasks. Our method achieves performance that is averagely 1,139.1% of the demonstration.

## 1 Introduction

Imitation learning (IL) offers an alternative to reinforcement learning (RL) for training an agent, which mimics the demonstrations of an expert and avoids manually designed reward functions. Behavioral cloning (BC) (Pomerleau, 1991) is the simplest form of imitation learning, which learns a policy using supervised learning. More advanced methods, inverse reinforcement learning (IRL) (Ng & Russell, 2000; Abbeel & Ng, 2004) seeks to recover a reward function from the demonstrations and train an RL agent on the recovered reward function. In the maximum entropy variant of IRL, the aim is to find a reward function that makes the demonstrations appear near-optimal on the principle of maximum entropy (Ziebart et al., 2008; 2010; Boularias et al., 2011; Finn et al., 2016).

However, most state-of-the-art IRL methods fail to meet the performance of demonstrations in high-dimensional environments with limited demonstration data, e.g., a one-life demonstration in Atari domain (Yu et al., 2020). This is due to the main goal of these IRL approaches is to recover a reward function that justifies the demonstrations only. The rewards recovered from limited demonstration data would be vulnerable to the overfitting problem. Optimizing these rewards from an arbitrary initial policy results in inferior performance. Recently, Yu et al. (2020) proposed generative intrinsic reward learning for imitation learning with limited demonstration data. This method outperforms expert and IRL methods in several Atari games. Although GIRIL uses the prediction error as curiosity to design the surrogate reward that encourages (pushes) states away from the demonstration and avoids overfitting, the curiosity also results in ambiguous quality of the rewards in the environment.

In this paper, we focus on addressing the two key issues of previous methods when learning with limited demonstration data, i.e., 1) overfitting problem, and 2) ambiguous quality of the reward function. To address these issues, we propose to learn a straightforward surrogate reward function by learning to plan from the demonstration data, which is more reasonable than the previous intrinsic reward function (i.e., the prediction error between states). Differential planning modules (DPM) is potentially useful to achieve this goal, since it learns to map observation to a planning computation for a task, and generates action predictions based on the resulting plan (Tamar et al., 2016; Nardelli et al., 2019; Zhang et al., 2020). Value iteration networks (VIN) (Tamar et al., 2016) is the representative one, which represents value iteration as a convolutional neural network (CNN). Meaningful reward and value maps have been learned along with the useful planning computation, which leads to policies that generalize well to new tasks. However, due to the inefficiency of summarizing complicated transition dynamics, VIN fails to scale up to the Atari domain.

To address this challenge, we propose a novel method called variational planning-embedded reward learning (vPERL), which is composed of two submodules: a planning-embedded action back-tracing module and the transition dynamics module. We leverage a variational objective based on the conditional variational autoencoder (VAE) (Sohn et al., 2015) to jointly optimize the two submodules, which greatly improves the generalization ability. This is critical for the success of achieving a straightforward and smooth reward function and value function with limited demonstration data.

As shown in Figure 1, vPERL learns meaningful reward and value maps that attends to the resulting region of the agent executing an action, which indicates meaningful planning computation. However, directly applying VIN in Atari domain in the way of supervised learning (Tamar et al., 2016) only learns reward and value maps that attend no specific region, which usually results in no avail.

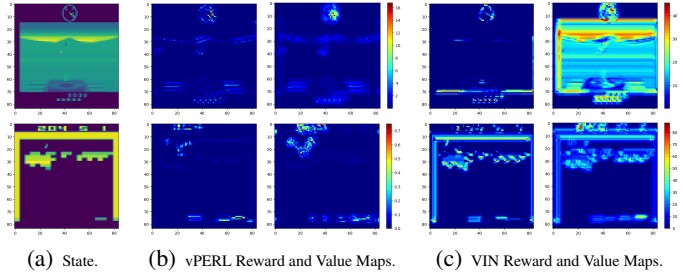

(a) State.     (b) vPERL Reward and Value Maps.     (c) VIN Reward and Value Maps.

Figure 1: Visualization of state, reward and value maps of vPERL and VIN on Battle Zone game (the first row) and Breakout game (the second row).

Empirical results show that our method outperforms state-of-the-art IRL methods on multiple Atari games and continuous control tasks. Remarkably, our methods achieve performance that is up to 58 times of the demonstration. Moreover, the average performance improvement of our method is 1,139.1% of the demonstration over eight Atari games.

## 2   BACKGROUND AND RELATED LITERATURE

**Markov Decision Process (MDP)**(Bellman, 1966) is a standard model for sequential decision making and planning. An MDP $M$ is defined by a tuple $(\mathcal{S}, \mathcal{A}, T, R, \gamma)$, where $\mathcal{S}$ is the set of states, $\mathcal{A}$ is the set of actions, $T : \mathcal{S} \times \mathcal{A} \times \mathcal{S} \to \mathbb{R}_+$ is the environment transition distribution, $R : \mathcal{S} \to \mathbb{R}$ is the reward function, and $\gamma \in (0, 1)$ is the discount factor (Puterman, 2014). The expected discounted return or *value* of the policy $\pi$ is given by $V^\pi(s) = \mathbb{E}_\tau[\sum_{t=0} \gamma^t R(s_t, a_t) | s_0 = s]$, where $\tau = (s_0, a_0, s_1, a_1, \cdots)$ denotes the trajectory, in which the actions are selected according to $\pi$, $s_0 \sim \mathbb{T}_0(s_0)$, $a_t \sim \pi(a_t | s_t)$, and $s_{t+1} \sim T(s_{t+1} | s_t, a_t)$. The goal in an MDP is to find the optimal policy $\pi^*$ that enables the agent to obtain high long-term rewards.

**Generative Adversarial Imitation Learning (GAIL)** (Ho & Ermon, 2016) extends IRL by integrating adversarial training technique for distribution matching (Goodfellow et al., 2014). GAIL performs well in low-dimensional applications, e.g., MuJoCo. However, it does not scale well to high-dimensional scenarios, such as Atari games (Brown et al., 2019a). **Variational adversarial imitation learning (VAIL)** (Peng et al., 2019) improves on GAIL by compressing the information via variational information bottleneck. GAIL and VAIL inherit problems of adversarial training, such as instability in training process, and are vulnerable to overfitting problem when learning with limited demonstration data. We have included both methods as comparisons to vPERL in our experiments.

**Generative Intrinsic Reward driven Imitation Learning (GIRIL)** (Yu et al., 2020) leverage generative model to learn generative intrinsic rewards for better exploration. Though GIRIL outperforms previous IRL methods on several Atari games, the reward map of GIRIL is ambiguous and less informative, which results in inconsistent performance improvements in different environments. In contrast, our vPERL learns efficient planning-based reward that is more straightforward and informative. We have included GIRIL as a competitive baseline in our experiments.

**Differentiable planning modules** perform end-to-end learning of planning computation, which leads to policies that generalize to new tasks. Value iteration (VI) (Bellman, 1957) is a well-known method for calculating the optimal *value* $V^*$ and optimal policy $\pi^*$: $V_{n+1}(s) = \max_a Q_n(s, a)$,

where $Q_n(s,a) = R(s,a) + \gamma \sum_{s'} T(s'|s,a)V_n(s')$ denotes the *Q value* in the $n$th iteration. The value function $V_n$ in VI converges as $n \to \infty$ to $V^*$, from which the optimal policy may be derived as $\pi^*(s) = \arg\max_a Q_\infty(s,a)$.

Value iteration networks (VIN) (Tamar et al., 2016) proposes to embed value iteration (VI) (Bellman, 1957) process with a recurrent convolutional network, and generalizes well in conventional navigation domains. VIN assumes there is some unknown embedded MDP $\overline{M}$ where the optimal plan in $\overline{M}$ contains useful information about the optimal plan in the original MDP $M$. VIN connects the two MDPs with a parametric reward function $\overline{R} = f_R(s)$. Nardelli et al. (2019) proposes value propagation networks (VPN) generalize VIN for better sample complexity by employing value propagation (VProp). Recently, universal value iteration networks (UVIN) extends VIN to spatially variant MDPs (Zhang et al., 2020). Although VIN can be extended to irregular spatial graphs by applying graph convolutional operator (Niu et al., 2018), most of the VIN variants still focus on solving the conventional navigation problems (Zhang et al., 2020).

In this paper, we extend differentiable planning module to learn an efficient reward function for imitation learning on limited demonstration data. We dig more on leveraging the learned reward function for imitation learning; while previous related work of VIN focuses more on the value function. Therefore, our work is complementary to the research of VIN and its variants. Note that any differentiable planning module can be embedded in our method. As a simple example, we utilize the basic VIN as a backbone to build our reward learning module.

## 3 VARIATIONAL PLANNING-EMBEDDED REWARD LEARNING

In this section, we introduce our solution, variational planning-embedded reward learning (vPERL). As illustrated in Figure 2, our reward learning module is composed of two submodules to accomplish planning-embedded action back-tracing and explicit forward transition dynamics modeling.

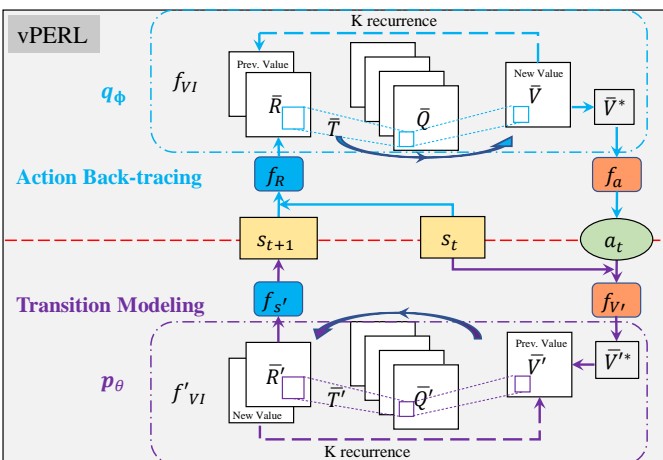

Figure 2: Illustration of the proposed vPERL model.

### 3.1 ACTION BACK-TRACING AND FORWARD DYNAMICS MODELLING IN VPERL

**Planning-embedded action back-tracing.** Instead of directly applying VIN for policy learning (Tamar et al., 2016), we build our first submodule $q_\phi(a_t|s_t, s_{t+1})$ for action back-tracing. As illustrated in the top section of Figure 2, we first obtain the reward map $\overline{R} = f_R(s_t, s_{t+1})$ on an embedded MDP $\overline{M}$, where $f_R$ is a convolutional layer. A VI module $f_{VI}$ takes in the reward map $\overline{R}$, and effectively performs $K$ times of VI by recurrently applying a convolutional layer $\overline{Q}$ for $K$ times (Tamar et al., 2016). The $\overline{Q}$ layer is then max-pooled to obtain the next-iteration value $\overline{V}$. The right-directed circular arrow in a light-blue color denotes the direction of convolutions. Then, we simply obtain the action from the intermediate optimal value $\overline{V}^*$ by an action mapping function: $a_t = f_a(\overline{V}^*)$. On these terms, we build our planning-embedded action back-tracing submodule, which is formally represented as $q_\phi(a_t|s_t, s_{t+1}) = f_a(f_{VI}(f_R(s_t, s_{t+1})))$. Since the

convolutional kernel is incapable of summarizing the transition dynamics in a complex environment, directly training this submodule is still insufficient for learning efficient reward function and planning computation in an environment like Atari domain.

**Explicit transition dynamics modeling via inverse VI.** To address this, we further build upon another submodule $p_\theta(s_{t+1}|a_t, s_t)$ for explicit transition dynamics modeling. We build the submodule based on the inverse VI module, which is a NN architecture that mimics the process of the inverse version of VI. The implementation of the inverse VI module is straightforward. We first map the action for the intermediate optima value in another embedded MDP $\overline{M'}$ by a value mapping function: $\overline{V'}^* = f_{V'}(s_t, a_t)$. Then, we apply the inverse VI module to obtain the reward map $\overline{R'}$. The inverse VI module $f'_{VI}$ takes in the intermediate value $\overline{V'}$ and recurrently apply a deconvolutional layer $\overline{Q'}$ for $K$ times on the value to obtain the reward map $\overline{R'}$. The left-directed circular arrow in a purple color denotes the direction of deconvolutions. To accomplish the transition, we map the obtained $\overline{R'}$ to the future state by: $s_{t+1} = f_{s'}(\overline{R'})$. The transition modeling is therefore presented as $p_\theta(s_{t+1}|a_t, s_t) = f_{s'}(f'_{VI}(f_{V'}(s_t, a_t)))$, which is a differentiable submodule, and can be trained simultaneously with the action back-tracing submodule.

**Variational solution to vPERL.** A variational autoencoder (VAE) (Kingma & Welling, 2013) can be defined as being an autoencoder whose training is regularised to avoid overfitting and ensure that the latent space has good properties that enable generative process. To avoid the learned planning-based reward overfitting to the demonstration, we optimize both submodules in a unified variational solution, which follows the formulation of conditional VAE (Sohn et al., 2015). Conditional VAE is a conditional generative model for structured output prediction using Gaussian latent variables, which is composed of a conditional encoder, decoder and prior. Accordingly, we regard the action back-tracing module $q_\phi(z|s_t, s_{t+1})$ as the encoder, $p_\theta(s_{t+1}|z, s_t)$ as the decoder, and $p_\theta(z|s_t)$ as the prior. Our vPERL module is maximized with the following objective:

$$\mathcal{L}(s_t, s_{t+1}; \theta, \phi) = \mathbb{E}_{q_\phi(z|s_t, s_{t+1})}[\log p_\theta(s_{t+1}|z, s_t)] - \mathrm{KL}(q_\phi(z|s_t, s_{t+1})\|p_\theta(z|s_t)) \\ - \alpha\mathrm{KL}(q_\phi(\hat{a}_t|s_t, s_{t+1})\|\pi_E(a_t|s_t)) \tag{1}$$

where $z$ is the latent variable, $\pi_E(a_t|s_t)$ is the expert policy distribution, $\hat{a}_t = \mathrm{Softmax}(z)$ is the transformed latent variable, $\alpha$ is a positive scaling weight. The first two terms on the RHS of Eq. (1) in the first line denote the evidence lower bound (ELBO) of the conditional VAE (Sohn et al., 2015). These two terms are critical for our reward learning module to perform planning-based action back-tracing and transition modeling. Additionally, we integrate the third term on the RHS of Eq. (1) in the second line to further boost the action back-tracing. The third term minimizes the KL divergence between the expert policy distribution $\pi_E(a_t|s_t)$ and the action distribution $q_\phi(\hat{a}_t|s_t, s_{t+1})$, where $\hat{a}_t = \mathrm{Softmax}(z)$ is transformed from the latent variable $z$. In this way, we train the forward state transition and action back-tracing simultaneously.

---

**Algorithm 1** Imitation learning via variational planning-embedded reward learning (vPERL).

1: **Input:** Expert demonstration data $\mathcal{D} = \{(s_i, a_i)\}_{i=1}^N$.
2: Initialize policy $\pi$, and the dual planning networks.
3: **for** $e = 1, \cdots, E$ **do**
4:     Sample a batch of demonstration $\tilde{\mathcal{D}} \sim \mathcal{D}$.
5:     Train vPERL module on $\tilde{\mathcal{D}}$ to converge.
6: **end for**
7: **for** $i = 1, \cdots, \mathrm{MAXITER}$ **do**
8:     Update policy via any policy gradient method, e.g., PPO on the learned surrogate reward $r_t$.
9: **end for**
10: **Output:** Policy $\pi$.

---

Note that the full objective in Eq. (1) is still a variational lower bound of the marginal likelihood $\log(p_\theta(s_{t+1}|s_t))$. Accordingly, it is reasonable to maximize this as an objective of our reward learning module. By optimizing the objective, we improve the forward state transition and action back-tracing. As a result, our reward learning module efficiently models the transition dynamics of the environment. During training, we use the latent variable $z$ as the intermediate action. After training, we will calculate the surrogate rewards from the learned reward map. As shown in Figure 1, our method learns meaningful reward map, which highlights the resulting region of an agent executing one action.

To leverage such meaningful information, we calculate two types of rewards that both correspond to the highlighted informative region, i.e., $r_t = R_{\text{Max}} = \max \overline{R}$ and $r_t = R_{\text{Mean}} = \text{mean} \overline{R}$, which uses the maximum and mean value of the reward map $\overline{R}$, respectively.

Algorithm 1 summarizes the full training procedure of imitation learning via vPERL. The process begins by training a vPERL module for $E$ epochs (steps 3-6). In each training epoch, we sample a mini-batch demonstration data $\tilde{\mathcal{D}}$ with a mini-batch size of $B$ and maximize the objective in Eq. (1). Then in steps 7-9, we update the policy $\pi$ via any policy gradient method, e.g., PPO (Schulman et al., 2017), so as to optimize the policy $\pi$ with the learned surrogate reward function $r_t$.

## 4 Experiments

### 4.1 Atari Games

We first evaluate our proposed vPERL on one-life demonstration data for eight Atari games within OpenAI Gym (Brockman et al., 2016). To enable a fair comparison, we evaluate our method and all other baselines under the same standard setup, where we train an agent to play Atari games without access to the true reward function (Ibarz et al., 2018; Brown et al., 2019a). The games and demonstration details are provided in Table 1.

A one-life demonstration only contains the states and actions performed by an expert player until they lose their life in a game for the first time (Yu et al., 2020). In contrast, one full-episode demonstration contains states and actions after the expert player loses all available lives in a game. Therefore, the one-life demonstration data is much more limited than one full-episode demonstration. We define three levels of performance: 1) basic one-life demonstration-level - gameplay up to one life lost ("one-life"), 2) expert-level - gameplay up to all-lives lost ("one full-episode"), and 3) beyond expert - "better-than-expert" performance.

Table 1: Statistics of Atari environments.

| Game | Demonstration Length | | # Lives available |
| --- | --- | --- | --- |
| | One-life | Full-episode | |
| Kung-Fu Master | 1,167 | 3,421 | 4 |
| Battle Zone | 260 | 1,738 | 5 |
| Centipede | 166 | 663 | 3 |
| Seaquest | 562 | 2,252 | 4 |
| Q*bert | 787 | 1,881 | 4 |
| Breakout | 1,577 | 2,301 | 5 |
| Beam Rider | 1,875 | 4,587 | 3 |
| Space Invaders | 697 | 750 | 3 |

Our ultimate goal is to train an imitation agent that can achieve a better-than-expert performance with the demonstration data recorded up to the moment of losing their first life in the game.

**Demonstrations** To generate one-life demonstrations, we trained a PPO (Schulman et al., 2017) agent with the ground-truth reward for 10 million simulation steps. We used PPO implementation with the default hyper-parameters in the repository (Kostrikov, 2018). As Table 1 shows, the one-life demonstrations are all much shorter than the full-episode demonstrations, which make for extremely limited training data.

**Experimental Setup** Our first step was to train a reward learning module for each game on the one-life demonstration. We set $K = 10$ in vPERL for all of the Atari games. By default, we use a neural network architecture that keeps the size of the reward map and value maps the same as that of the state, which is $84 \times 84$. We achieve this by using a convolutional kernel of size 3 for each convolutional layer, and applying padding. The corresponding method is called 'vPERL-Large'. Additionally, to enable faster learning, we implement our method with another neural network architecture that reduces the size of the reward map and value maps into $18 \times 18$. The corresponding method is called 'vPERL-Small'. Both vPERL-Large and vPERL-Small can learn meaningful reward map as well as useful planning computation. Training was conducted with the Adam optimizer (Kingma & Ba, 2015) at a learning rate of 3e-5 and a mini-batch size of 32 for 50,000 epochs. In each training epoch, we sampled a mini-batch of data every four states.

To evaluate the quality of our learned reward, we trained a policy to maximize the inferred reward function via PPO. We set $\alpha = 100$ for training our reward learning module. We trained the PPO on the learned reward function for 50 million simulation steps to obtain our final policy. The PPO is trained with a learning rate of 2.5e-4, a discount factor of 0.99, a clipping threshold of 0.1, an entropy coefficient of 0.01, a value function coefficient of 0.5, and a GAE parameter of 0.95 (Schulman et al.,

2016). We compared imitation performance by our vPERL agent against VIN, two state-of-the-art inverse reinforcement learning methods, GAIL (Ho & Ermon, 2016) and VAIL (Peng et al., 2019). More details of setup are outlined in Appendix F.2.

**Results** In Figure 3, we report the performance by normalizing the demonstration performance to 1. Figure 3 shows that vPERL achieves performance that is usually close or better than that of the expert demonstrator. The most impressive one is the Centipede game, our vPERL achieves performance that is around 60 times higher than the demonstration. GIRIL achieves the second best performance in Centipede, beating the demonstration by around 30 times. On the Qbert game, vPERL beats all other baselines by a large margin, achieving performance that is more than 15 times of the demonstration.

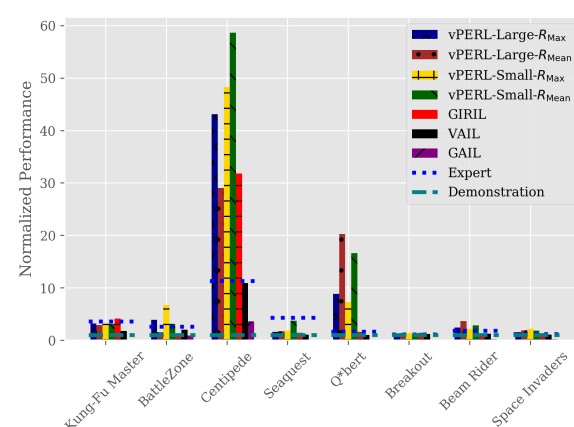

Figure 3: Performance improvement of vPERL and baselines.

Table 2: Average return of vPERL with two types network architectures (Large and Small) and two types of rewards ($R_{\mathrm{Max}}$ and $R_{\mathrm{Mean}}$), GIRIL (Yu et al., 2020), VIN and state-of-the-art IRL algorithms GAIL (Ho & Ermon, 2016) and VAIL (Peng et al., 2019) with one-life demonstration data on eight Atari games. The results shown are the mean performance over five random seeds with better-than-expert performance in bold. The last two rows show the average performance improvements of IL algorithms versus demonstration (Demo.) and expert performance with greater than 100% in bold.

| Game | Expert Average | Demo. Average | PPO-50M Average | vPERL-Large $R_{\mathrm{Max}}$ | vPERL-Large $R_{\mathrm{Mean}}$ | vPERL-Small $R_{\mathrm{Max}}$ | vPERL-Small $R_{\mathrm{Mean}}$ | GIRIL | VAIL | GAIL | VIN |
|---|---|---|---|---|---|---|---|---|---|---|---|
| Kung-Fu Master | 23,434.8 | 6,500.0 | 38,580.0 | 20,700.0 | 18,900.0 | 20,800.0 | 20,518.3 | **26,742.6** | 11,751.2 | 1.3 | 190.4 |
| Battle Zone | 18,137.5 | 7,000.0 | 33,400.0 | **27,500.0** | 11,000.0 | **47,200.0** | **21,899.1** | 7,372.8 | 14,222.7 | 6,200.0 | 3,372.0 |
| Centipede | 4,069.4 | 360.0 | 7,704.6 | **15,516.9** | **10,448.8** | **17,371.4** | **21,113.4** | **11,446.5** | 3,936.1 | 1,306.5 | 1,907.6 |
| Seaquest | 1,744.5 | 440.0 | 1,920.0 | 706.0 | 756.0 | 816.0 | 1,634.2 | 624.8 | 476.1 | 60.0 | 148.0 |
| Q*bert | 13,441.5 | 8,150.0 | 26,235.0 | **71,920.0** | **165,325.0** | **58,965.0** | **135,598.0** | 13,070.3 | 8,314.8 | 25.0 | 283.4 |
| Breakout | 346.4 | 305.0 | 522.9 | **374.7** | **420.0** | **422.3** | **348.8** | **430.3** | **370.6** | 2.0 | 0.4 |
| Beam Rider | 2,447.7 | 1,332.0 | 5,146.2 | **3,264.6** | **4,890.0** | **2,960.5** | **3,801.8** | 1,599.5 | 1,710.8 | 286.5 | 351.4 |
| Space Invaders | 734.1 | 600.0 | 2,880.5 | **915.2** | **1,135.0** | **1,325.5** | **1,106.5** | 556.6 | 646.2 | 113.9 | 227.4 |
| Average | 345.8% | 100.0% | - | **823.1%** | **780.5%** | **913.8%** | **1,139.1%** | **544.0%** | **255.6%** | 63.3% | 85.3% |
| Improvements (%) | 100.0% | 47.8% | - | **194.9%** | **267.9%** | **210.1%** | **278.5%** | **103.9%** | 72.1% | 12.2% | 15.2% |

A detailed quantitative comparison of IL algorithms is listed in Table 2. We have evaluated four variants of vPERL with two types of network architectures (Large and Small) and surrogate rewards ($R_{\mathrm{Max}}$ and $R_{\mathrm{Mean}}$). Both vPERL-Large and vPERL-Small can learn meaningful reward and value maps as well as useful computation in the Atari domain. In Appendix D and E, we visualize the learned reward and value maps of vPERL-Large and vPERL-Small, respectively. With such meaningful rewards, the four variants of vPERL outperform the expert demonstrator in six out of eight Atari games. Remarkably, vPERL-Small with $R_{\mathrm{Mean}}$ achieves an average performance that is 1,139.1% of the demonstration and 278.5% of the expert over the eight Atari games. Figure 3 shows the bar plot of normalized performance of four vPERL variants against other baselines.

Table 2 shows that VIN is far from achieving demonstration-level performance, since it is unable to learn useful planning computation as well as the meaningful reward and value maps in the Atari domain. GAIL fails to achieve good imitation learning performance. VAIL manages to exceed the expert performance in one game, i.e., Breakout. GIRIL performs better than previous methods, outperforming the expert demonstrator in three games. The results show that our vPERL agent outperforms the expert demonstrator by a large margin in six out of eight games. Figure 4 shows the qualitative comparison of our method (vPERL-Small-$R_{\mathrm{Mean}}$), GIRIL, VAIL, GAIL, VIN and the average performance of the expert and demonstration. Additionally in Appendix C.1, our method consistently outperforms expert and other baselines on two more Atari games, Krull and Time Pilot.

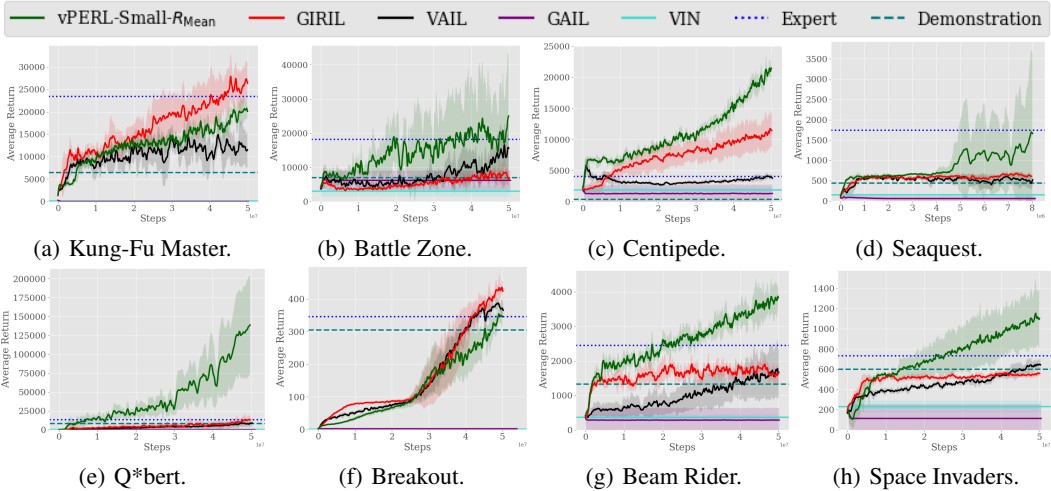

Figure 4: Average return vs. the number of simulation steps on Atari games. The solid lines show the mean performance over five random seeds. The shaded area represents the standard deviation from the mean. The blue dotted line denotes the average return of the expert. The area above the blue dotted line indicates performance beyond the expert.

#### 4.1.1 HOW DOES VPERL OUTPERFORM PREVIOUS METHODS?

**The contributions of each component of vPERL.** In this subsection, we study the contribution of each component of our method, i.e. Action Back-tracing submodule, Transition Modeling submodule, and the variational objective in Eq. (1). Specifically, we directly train the Action Back-tracing and Transition Modeling submodules in terms of supervised learning. We used the mean of the reward map and the prediction error of the next state as the reward for the former and latter submodules, respectively. To study the contribution of the variational objective, we introduced another baseline, PERL, which trains both submodules as an autoencoder. Table 3 shows quantitative comparison between the performance of vPERL and its components.

Table 3: Average return of vPERL-Small-$R_{\mathrm{Mean}}$, and its components (i.e., Action Back-tracing and Transition Modeling, and PERL) with one-life demonstration data on eight Atari games. The results shown are the mean performance over five random seeds with better-than-expert performance in bold.

| | Expert | Demo. | vPERL-Small-$R_{\mathrm{Mean}}$ | Components of vPERL-Small-$R_{\mathrm{Mean}}$ | | | |
| --- | --- | --- | --- | --- | --- | --- | --- |
| Game | Average | Average | Average | Action Back-tracing | Transition Modeling | PERL | Supervised PERL |
| Kung-Fu Master | 23,434.8 | 6,500.0 | 20,518.3 | 0.0 | 0.0 | 0.0 | 0.0 |
| Battle Zone | 18,137.5 | 7,000.0 | **21,899.1** | 6000.0 | 0.0 | 0.0 | 2,000.0 |
| Centipede | 4,069.4 | 360.0 | 21,113.4 | 151.0 | 2,216.7 | **16,479.6** | 2,485.1 |
| Seaquest | 1,744.5 | 440.0 | 1,634.2 | 0.0 | 0.0 | 140.0 | 80.0 |
| Q*bert | 13,441.5 | 8,150.0 | **135,598.0** | 74.9 | 150.0 | 0.0 | 0.0 |
| Breakout | 346.4 | 305.0 | **348.8** | 2.5 | 5.5 | 2.2 | 2.7 |
| Beam Rider | 2,447.7 | 1,332.0 | **3,801.8** | 22.0 | 784.8 | 17.6 | 0.0 |
| Space Invaders | 734.1 | 600.0 | **1,106.5** | 285.0 | 145.5 | 285.0 | 0.0 |

The results show that individual training of each component results in no avail. PERL successfully outperforms the demonstration in one game, i.e. Centipede, which indicates the potential advantage of using both submodules. However, PERL fails in the other seven games, while vPERL outperforms the demonstration in eight games and outperforms expert in six. The large performance gap between PERL and vPERL indicates the variational objective in Eq. (1) is important to learn efficient rewards. To further investigate the key reason for why our method works well, we added another baseline - supervised PERL, which forces the encoding of PERL to be close to the true action. The supervised PERL fails in all of the games. Comparing with vPERL, we can attribute the critical contribution to the use of the ELBO of conditional VAE, or more specific, the term $\mathrm{KL}(q_\phi(z|s_t, s_{t+1}) \| p_\theta(z|s_t))$ in Eq. (1). It helps vPERL to work well and outperform previous methods for two reasons:
**1)** The generative training of VAE can serve as a good regularization to alleviate the overfitting problem.

**2)** The regularization enables vPERL to learn a smooth value function and reward function, which consistently provides straightforward and informative rewards for the moving states in the environment.

**Empirical evidence:**

**1) Better generalization ability.** The empirical results in Table 2 and Figure 4 show that VIN, GAIL and VAIL are vulnerable to overfitting problem, usually results in no avail and has fewer chances to reach the demonstration-level performance. In contrast, our vPERL has better generalization ability and consistently achieves performance that is either close to or better than the expert.

**2) Straightforward and informative reward.** Figure 5 shows the state, the reward maps of vPERL and GIRIL in three Atari games. The reward map of GIRIL can be close to zero (in Battle Zone) and state (in Q*bert) or occasionally informative (in Centipede), which is ambiguous and less informative. In contrast, the reward map of our vPERL is more straightforward, and consistently attends to informative regions in the state for all of the games.

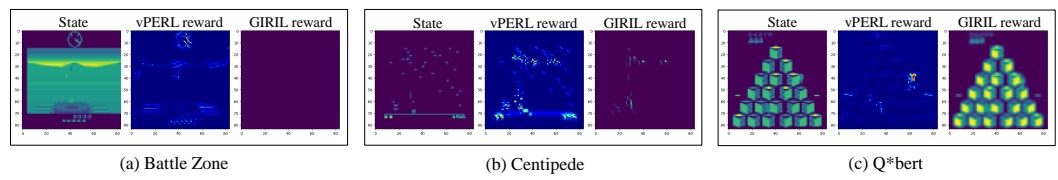

(a) Battle Zone         (b) Centipede         (c) Q*bert

Figure 5: Visualization of state, vPERL reward map and GIRIL reward map for Atari games.

Our method successfully addresses the two key issues, therefore, it can outperform previous methods.

## 4.2 CONTINUOUS CONTROL TASKS

We also evaluated our method on continuous control tasks where the state space is low-dimensional and the action space is continuous. The continuous control tasks were from Pybullet [1] environment.

**Demonstrations** To generate demonstrations, we trained a Proximal Policy Optimization (PPO) agent with the ground-truth reward for 1 million simulation steps. We used the PPO implementation in the repository (Kostrikov, 2018) with the default hyper-parameters for continuous control tasks. In each task, we used one demonstration with a fixed length of 1,000 for evaluation. The details of experimental setup can be found in Appendix F.1.

**Results** Figure 6 shows that our method vPERL achieves the best imitation performance in both continuous control tasks, i.e. Inverted Pendulum and Inverted Double Pendulum. Although GIRIL achieves performance that is close to the demonstration, the efficient planning-based reward function enables vPERL to perform even better. Other baselines are unable to reach the demonstration-level performance by learning from only one demonstration. Quantitative results are shown in Appendix A.

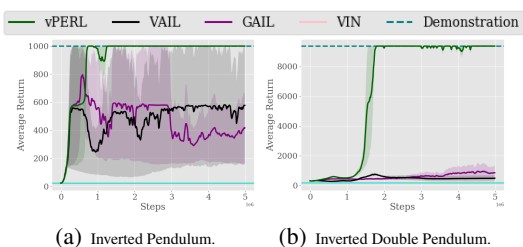

(a) Inverted Pendulum.    (b) Inverted Double Pendulum.

Figure 6: Average return vs. number of simulation steps on continuous control tasks.

## 4.3 ABLATION STUDIES

**Ablation study on the growing number of demonstrations.** Figure 7 shows the average return versus number of full-episode demonstrations on both Atari games and continuous control tasks. The results shows that our method vPERL achieves the highest performance across different numbers of full-episode demonstrations. GIRIL usually comes the second best, and GAIL can achieve good performance with more demonstrations in continuous control tasks. Quantitative results have been shown in Appendix B.1.

---

[1]https://pybullet.org/

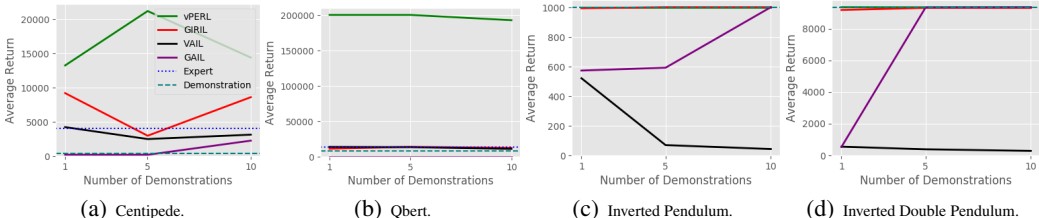

(a) Centipede.  (b) Qbert.  (c) Inverted Pendulum.  (d) Inverted Double Pendulum.

Figure 7: Average return vs. number of demonstrations on Atari games and continuous control tasks.

**Ablation study on the optimality of the demonstrations.** Figure 8 shows the average return versus optimality of demonstrations on Atari games and continuous control tasks. In sections 4.1 and 4.2, we trained PPO agents with ground-truth reward for 10 million (10M) steps as the expert for Atari games, and for 1 million (1M) steps as the expert for continuous control tasks. In this ablation, we train PPO agents with 10% and 50% simulation steps of the expert to generate demonstrations with diverse optimality. The results show that vPERL consistently outperforms the expert and demonstrations on the demonstrations of different optimality. Quantitative results are shown in Appendix B.2.

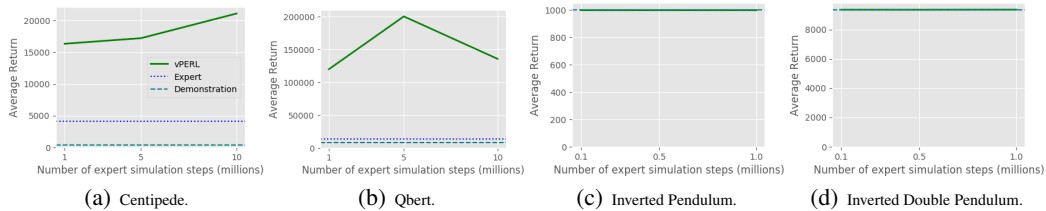

(a) Centipede.  (b) Qbert.  (c) Inverted Pendulum.  (d) Inverted Double Pendulum.

Figure 8: Average return vs. number of the expert simulation steps ($N_E$) on Atari games and continuous control tasks.

**Ablation study on the hyper-parameters $K$.** Figure 8 shows the average return versus different choices of $K$ on Atari games and continuous control tasks. The results show that our method is not very sensitive to different choices of $K$. Quantitative results are shown in Appendix B.3.

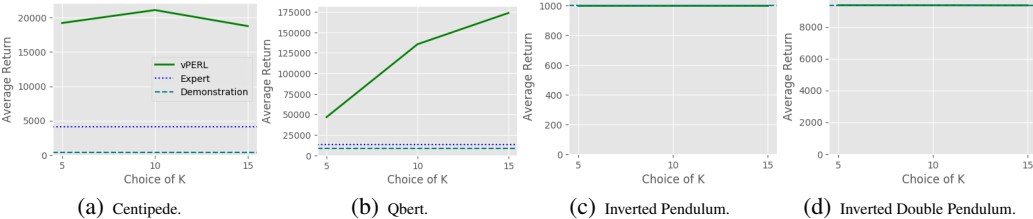

(a) Centipede.  (b) Qbert.  (c) Inverted Pendulum.  (d) Inverted Double Pendulum.

Figure 9: Average return vs. choices of $K$ on Atari games and continuous control tasks.

## 5 CONCLUSION

This paper presents a simple but efficient reward learning method, called variational planning-embedded reward learning (vPERL). By simultaneously training a planning-embedded action back-tracing module and a transition dynamics module in a unified generative solution, we obtain a reward function that is straightforward and informative, and has better generalization ability than previous methods. Informative analysis and empirical evidence support the critical contribution of ELBO regularization term for learning efficient planning-based reward with extremely limited demonstrations. Empirical results show our method outperforms state-of-the-art imitation learning methods on multiple Atari games and continuous control tasks by a large margin. Extensive ablation studies show that our method is not very sensitive to the number of demonstrations, optimality of demonstration, and choices of the hyperparameter $K$. We remain the extension of our method to more complex continuous control tasks as future work. Another interesting topic for future investigation would be applying vPERL to hard exploration tasks with extremely sparse rewards.

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

## A QUANTITATIVE RESULTS OF CONTINUOUS CONTROL TASKS.

Table 4 shows the detailed quantitative comparison of the demonstration and imitation methods. The results shown in the table were the mean performance over three random seeds.

Table 4: Average return of vPERL, GIRIL, VIN and state-of-the-arts inverse reinforcement learning algorithms GAIL (Ho & Ermon, 2016) and VAIL (Peng et al., 2019) with one demonstration data on continuous control tasks. The results shown are the mean performance over 3 random seeds with best imitation performance in bold.

| | Demonstration | vPERL | Imitation Learning Algorithms | | | |
|---|---|---|---|---|---|---|
| Task | Average | $R_{\text{Mean}}$ | GIRIL | VAIL | GAIL | VIN |
| Inverted Pendulum | 1,000.0 | **1,000.0** | 993.2 | 520.1 | 572.2 | 21.0 |
| Inverted Double Pendulum | 9,355.1 | **9,357.8** | 9,184.5 | 539.8 | 519.6 | 181.1 |

## B ABLATION STUDIES

### B.1 THE EFFECT OF THE NUMBER OF FULL-EPISODE DEMONSTRATIONS.

We also evaluated our method with different numbers of full-episode demonstrations on both Atari games and continuous control tasks. Table 5 and Table 6 show the detailed quantitative comparison of imitation learning methods across different numbers of full-episode demonstrations in the Centipede game and Qbert game. The comparisons on two continuous control tasks, Inverted Pendulum and Inverted Double Pendulum, have been shown in Table 7 and Table 8.

The results shows that our method vPERL achieves the highest performance across different numbers of full-episode demonstrations, and GIRIL usually comes the second best. GAIL is able to achieve better performance with the increase of the demonstration number in both continuous control tasks.

Table 5: Parameter Analysis of the vPERL versus other baselines with different numbers of full-episode demonstrations on Centipede game. The results shown are the mean performance over 5 random seeds with best performance in bold.

| # Demonstrations | vPERL | GIRIL | VAIL | GAIL |
|---|---|---|---|---|
| 1 | **13,253.2** | 9,195.1 | 4,212.8 | 171.0 |
| 5 | **21,219.2** | 2,952.5 | 2,465.6 | 161.0 |
| 10 | **14,406.2** | 8,599.6 | 3,108.0 | 2,232.4 |

Table 6: Parameter Analysis of the vPERL versus other baselines with different numbers of full-episode demonstrations on Qbert game. The results shown are the mean performance over 5 random seeds with best performance in bold.

| # Demonstrations | vPERL | GIRIL | VAIL | GAIL |
|---|---|---|---|---|
| 1 | **200,455.0** | 11,105.0 | 13,695.0 | 0.0 |
| 5 | **200,450.0** | **13,740.0** | 13,300.0 | 0.0 |
| 10 | **192,962.5** | 12,327.5 | 10,852.5 | 0.0 |

Table 7: Parameter Analysis of the vPERL versus other baselines with different numbers of full-episode demonstrations on Inverted Pendulum task. The results shown are the mean performance over 5 random seeds with best performance in bold.

| # Demonstrations | vPERL | GIRIL | VAIL | GAIL |
|---|---|---|---|---|
| 1 | **1,000.0** | 993.2 | 520.1 | 572.2 |
| 5 | **1,000.0** | 1,000.0 | 68.2 | 591.2 |
| 10 | **1,000.0** | 1,000.0 | 41.8 | 1,000.0 |

Table 8: Parameter Analysis of the vPERL versus other baselines with different numbers of full-episode demonstrations on InvertedDoublePendulum task. The results shown are the mean performance over 5 random seeds with best performance in bold.

| # Demonstrations | vPERL | GIRIL | VAIL | GAIL |
|---|---|---|---|---|
| 1 | **9,357.8** | 9,184.5 | 539.8 | 519.6 |
| 5 | **9,352.4** | 9,317.2 | 372.5 | 9,342.5 |
| 10 | **9,353.1** | 9,314.1 | 274.6 | 9,352.1 |

## B.2 THE EFFECT OF EXPERT OPTIMALITY.

Table 9 and Table 10 show the average return of vPERL-Small-$R_{\text{Mean}}$ with demonstrations of different optimality on Atari games and continuous control tasks, respectively. In the experiments, we trained a PPO agent with ground-truth reward for 10 million (10M) simulation steps as the expert for Atari games, and 1 million (1M) steps for continuous control tasks. To study the effects of optimality of the demonstrations, we additionally trained PPO agents with less simulation steps: 1M steps and 5M steps for Atari games, and 0.1M steps and 0.5M steps for continuous control tasks. With the additional PPO agents, we generated demonstrations with 10%, 50% optimality of the 10M-step 'Expert' for both Atari games and continuous control tasks.

The results show that our method outperforms expert by a large margin in Atari games and reach the demonstration-level performance in continuous control tasks with demonstrations of different optimality.

Table 9: Average return of vPERL-Small-$R_{\text{Mean}}$ with one-life demonstration data of expert policy trained under different simulations steps ($N_E$=1 million, 5 million and 10 million). The results shown are the mean performance over five random seeds with better-than-expert performance in bold.

| | Expert | Demonstration | | vPERL-Small-$R_{\text{Mean}}$ | |
|---|---|---|---|---|---|
| Game | Average | Average | $N_E$=1M | $N_E$=5M | $N_E$=10M |
| Battle Zone | 18,137.5 | 7,000.0 | **24,800.0** | **34,897.2** | **21,899.1** |
| Centipede | 4,069.4 | 360.0 | **16,343.7** | **17,223.3** | **21,113.4** |
| Q*bert | 13,441.5 | 8,150.0 | **147,565.5** | **200,300.0** | **135,598.0** |
| Beam Rider | 2,447.7 | 1,332.0 | **3,374.4** | **3,256.0** | **3,801.8** |

Table 10: Average return of vPERL-$R_{\text{Mean}}$ with one demonstration data of expert policy trained under different simulations steps ($N_E$=0.1 million, 0.5 million and 1 million). The results shown are the mean performance over five random seeds with demonstration-level performance in bold.

| | Demonstration | | vPERL-$R_{\text{Mean}}$ | |
|---|---|---|---|---|
| Game | Average | $N_E$=0.1M | $N_E$=0.5M | $N_E$=1M |
| Inverted Pendulum | 1,000 | **1,000.0** | **1,000.0** | **1,000.0** |
| Inverted Double Pendulum | 9,355.1 | **9,351.9** | **9,350.8** | **9,357.8** |

## B.3 THE EFFECT OF CHOICES OF $K$.

For the sake of consistency, we set $K = 10$ for all of experiments on Atari games and continuous control tasks in Section 4. To study the effects of the hyperparameter $K$, we evaluate our method on two Atari games and two continuous control tasks with two additional $K$ ($K$=5, and $K$=15).

Table 11 and Table 12 shows the average return of vPERL-Small-$R_{\text{Mean}}$ versus different choices of $K$ on Atari games and continuous control tasks. With the three choices of $K$, our method consistently outperforms the expert in the Atari games, and reach the best (demonstration-level) performance in continuous control tasks. This indicates that our method is not very sensitive to the choices of hyperparameter $K$.

Table 11: Average return of vPERL-Small-$R_{\mathrm{Mean}}$ with different choices of $K$, on one-life demonstration data. The results shown are the mean performance over five random seeds with better-than-expert performance in bold.

| Game | Expert Average | Demo. Average | vPERL-Small-$R_{\mathrm{Mean}}$ | | |
| --- | --- | --- | --- | --- | --- |
| | | | $K$=5 | $K$=10 | $K$=15 |
| Centipede | 4,069.4 | 360.0 | **19,228.8** | **21,113.4** | **18,776.2** |
| Q*bert | 13,441.5 | 8,150.0 | **46,705.5** | **135,598.0** | **173,735.0** |

Table 12: Average return of vPERL-$R_{\mathrm{Mean}}$ with different choices of $K$, on one-life demonstration data. The results shown are the mean performance over five random seeds with demonstration-level performance in bold.

| Game | Demonstration Average | vPERL-$R_{\mathrm{Mean}}$ | | |
| --- | --- | --- | --- | --- |
| | | $K$=5 | $K$=10 | $K$=15 |
| Inverted Pendulum | 1,000 | **1,000.0** | **1,000.0** | **1,000.0** |
| Inverted Double Pendulum | 9,355.1 | **9,358.8** | **9,357.8** | **9,351.9** |

## C  ADDITIONAL EVALUATION RESULTS

### C.1  ADDITIONAL ATARI GAMES.

Table 13 shows the average return of vPERL-Small with $R_{\mathrm{Mean}}$ and other baselines on two additional Atari games, Krull and Time Pilot. The results show that our method outperforms the expert and other baselines by a large margin on both additional Atari games.

Table 13: Average return of vPERL-Small with $R_{\mathrm{Mean}}$, GIRIL (Yu et al., 2020), VIN and state-of-the-art IRL algorithms GAIL (Ho & Ermon, 2016) and VAIL (Peng et al., 2019) with one-life demonstration data on additional Atari games. The results shown are the mean performance over five random seeds with better-than-expert performance in bold.

| Game | Expert Average | Demo. Average | vPERL-Small $R_{\mathrm{Mean}}$ | Imitation Learning Algorithms | | | |
| --- | --- | --- | --- | --- | --- | --- | --- |
| | | | | GIRIL | VAIL | GAIL | VIN |
| Krull | 8,262.0 | 2,826.0 | **19,412.5** | 6,515.0 | 5,819.7 | 264.0 | 2,013.3 |
| Time Pilot | 4,200.0 | 100.0 | **4,400.0** | 3,680.0 | 2,080.0 | 400.0 | 2,572.0 |

### C.2  ONE-LIFE DEMONSTRATIONS WITHOUT SCORES AND LIVES ON THE STATES.

To avoid the effects of the scores and lives in the states of Atari games, we also evaluate our method on the "No-score Demo.", which is obtained by masking the game score and number of lives left in the demonstrations (Brown et al., 2019a). Table 14 compares the average return of vPERL-Small-$R_{\mathrm{Mean}}$ with the "Standard Demo." and the "No-score Demo." on Q*bert game and Krull game.

Table 14: Average return of vPERL-Small-$R_{\mathrm{Mean}}$ with different choices of $K$ on one-life demonstration data. The results shown are the mean performance over five random seeds with better-than-expert performance in bold.

| Game | Expert Average | Demo. Average | vPERL-Small-$R_{\mathrm{Mean}}$ | |
| --- | --- | --- | --- | --- |
| | | | Standard Demo. | No-score Demo. |
| Krull | 8,262.0 | 2,826.0 | **19,412.5** | **27,009.6** |
| Q*bert | 13,441.5 | 8,150.0 | **135,598.0** | **200,500.0** |

The results show that our method achieves better performance on the "No-score Demo." than the "Standard Demo.". This indicates the negative effects of the game scores and numbers of left lives on the states of demonstrations. From Figure 1 and more reward visualization in Section D and E, we observe that our method learns to attend on the meaningful region in a state and ignore the game score and numbers of left lives automatically. Masking the game score and numbers of left

lives in the demonstration further alleviates burdens on learning efficient planning computations and planning-based rewards for Atari games.

In summary, our method can learn to outperform the expert without explicitly access to the true rewards, and does not relied on the game scores and numbers of left lives in the states of demonstrations. Furthermore, the results show that the performance of our method can be improved by masking out the game scores and numbers of left lives in the demonstrations.

## D   VISUALIZATION OF REWARD AND VALUE IMAGES OF VPERL-LARGE AND VIN.

In this section, we visualize the reward maps and value maps learned by vPERL-Large and VIN on Atari games. Here, both vPERL and VIN are based on large-size VIN architecture. The size of reward map is $84 \times 84$. The figures show that the reward and value maps learned by vPERL are much meaningful than that by VIN.

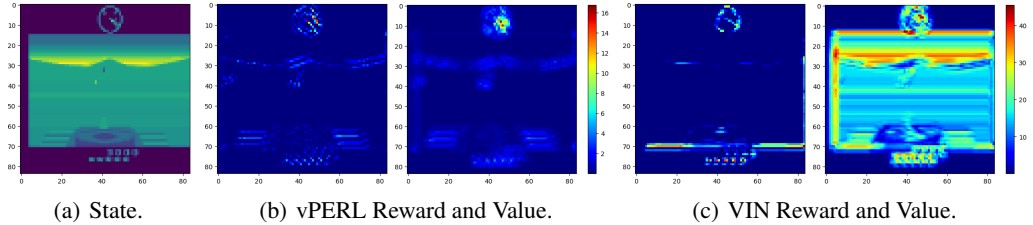

    (a) State.        (b)  vPERL Reward and Value.        (c)  VIN Reward and Value.

Figure 10: Visualization of state, reward map and value map on Battle Zone game. (a) The state, (b) the reward map and value map of vPERL, and (c) the reward map and value map of VIN.

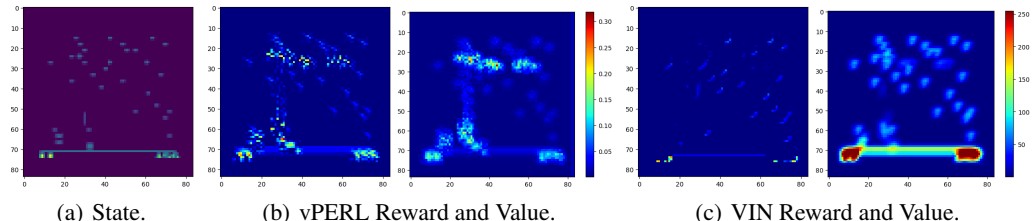

    (a) State.        (b)  vPERL Reward and Value.        (c)  VIN Reward and Value.

Figure 11: Visualization of state, reward map and value map on Centipede game. (a) The state, (b) the reward map and value map of vPERL, and (c) the reward map and value map of VIN.

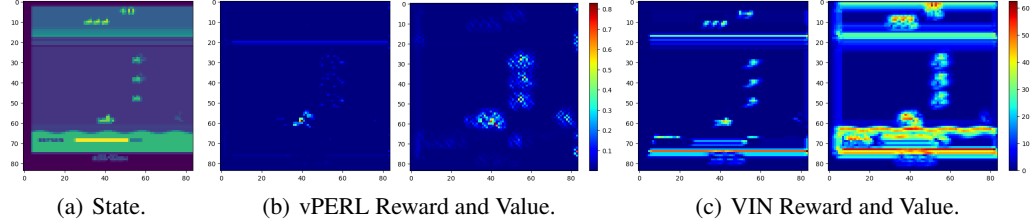

    (a) State.        (b)  vPERL Reward and Value.        (c)  VIN Reward and Value.

Figure 12: Visualization of state, reward map and value map on Seaquest game. (a) The state, (b) the reward map and value map of vPERL, and (c) the reward map and value map of VIN.

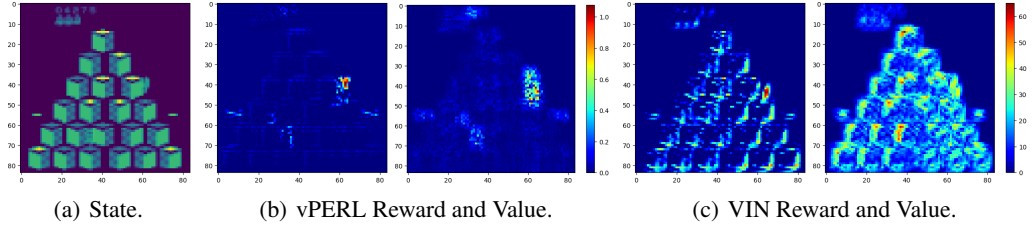

(a) State.  (b) vPERL Reward and Value.  (c) VIN Reward and Value.

Figure 13: Visualization of state, reward map and value map on Qbert game. (a) The state, (b) the reward map and value map of vPERL, and (c) the reward map and value map of VIN.

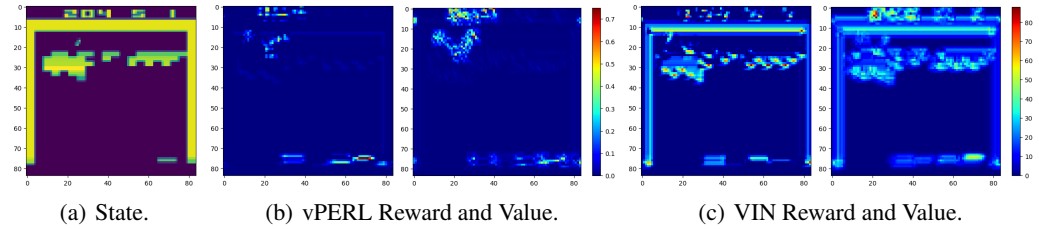

(a) State.  (b) vPERL Reward and Value.  (c) VIN Reward and Value.

Figure 14: Visualization of state, reward map and value map on Breakout game. (a) The state, (b) the reward map and value map of vPERL, and (c) the reward map and value map of VIN.

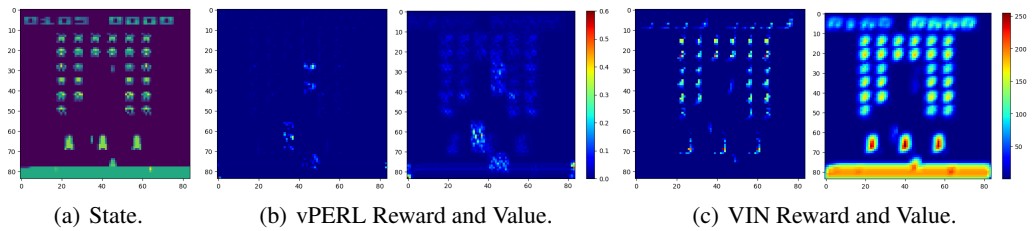

(a) State.  (b) vPERL Reward and Value.  (c) VIN Reward and Value.

Figure 15: Visualization of state, reward map and value map on Space Invaders game. (a) The state, (b) the reward map and value map of vPERL, and (c) the reward map and value map of VIN.

# E VISUALIZATION OF REWARD AND VALUE IMAGES OF VPERL-SMALL AND VIN.

In this section, we visualize the reward maps and value maps learned by vPERL and VIN on several Atari games. To enable faster training, here both vPERL and VIN are based on small-size VIN architecture. The size of reward map is $18 \times 18$. The figures show that the reward and value maps learned by vPERL are much meaningful than that by VIN.

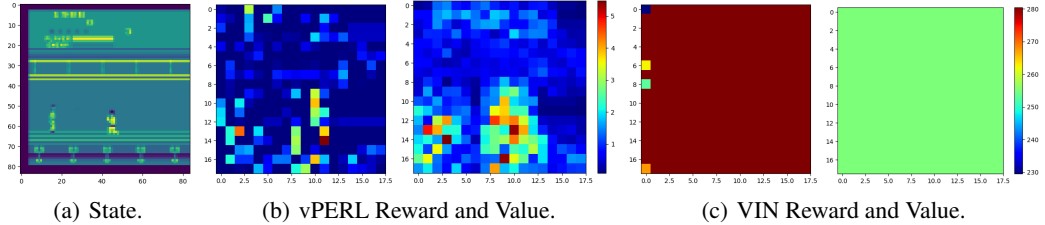

(a) State.  (b) vPERL Reward and Value.  (c) VIN Reward and Value.

Figure 16: Visualization of state, reward map and value map on Kung-Fu Master game. (a) The state, (b) the reward map and value map of vPERL, and (c) the reward map and value map of VIN.

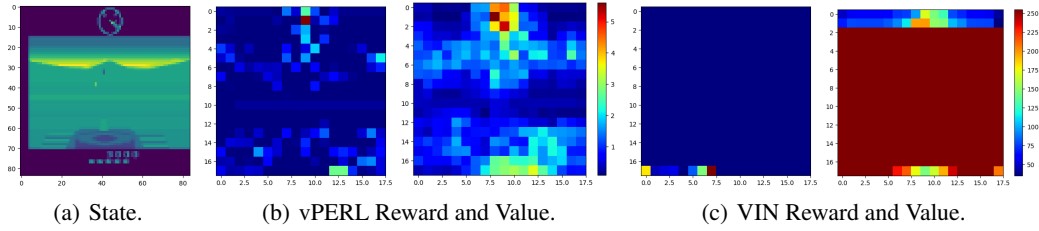

(a) State.  (b) vPERL Reward and Value.  (c) VIN Reward and Value.

Figure 17: Visualization of state, reward map and value map on Battle Zone game. (a) The state, (b) the reward map and value map of vPERL, and (c) the reward map and value map of VIN.

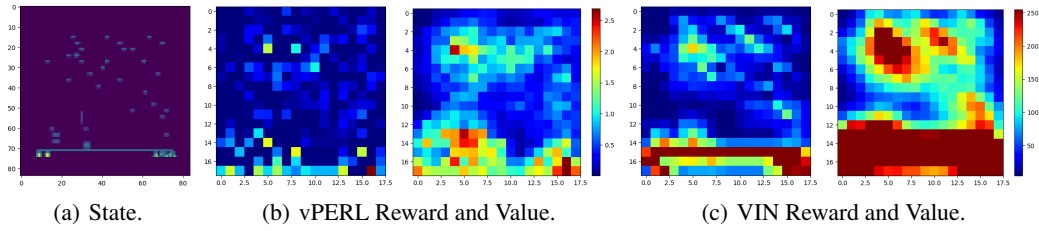

(a) State.  (b) vPERL Reward and Value.  (c) VIN Reward and Value.

Figure 18: Visualization of state, reward map and value map on Centipede game. (a) The state, (b) the reward map and value map of vPERL, and (c) the reward map and value map of VIN.

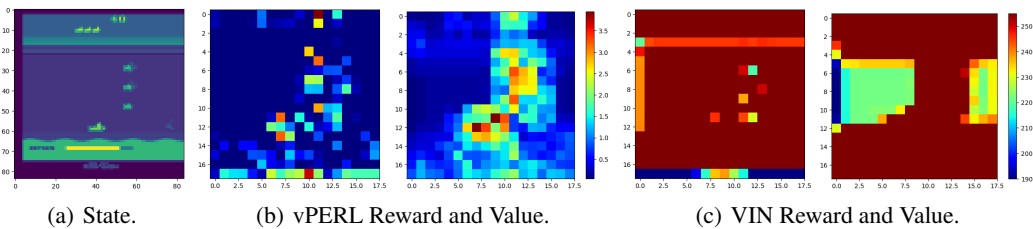

(a) State.  (b) vPERL Reward and Value.  (c) VIN Reward and Value.

Figure 19: Visualization of state, reward map and value map on Seaquest game. (a) The state, (b) the reward map and value map of vPERL, and (c) the reward map and value map of VIN.

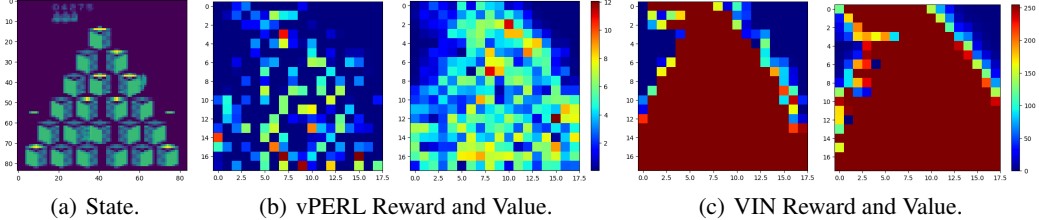

(a) State.  (b) vPERL Reward and Value.  (c) VIN Reward and Value.

Figure 20: Visualization of state, reward map and value map on Qbert game. (a) The state, (b) the reward map and value map of vPERL, and (c) the reward map and value map of VIN.

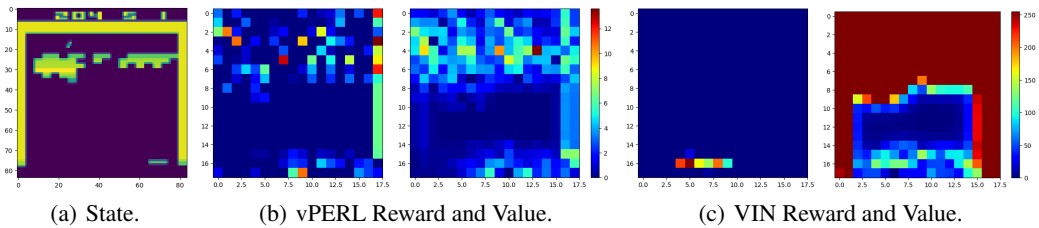

(a) State.  (b) vPERL Reward and Value.  (c) VIN Reward and Value.

Figure 21: Visualization of state, reward map and value map on Breakout game. (a) The state, (b) the reward map and value map of vPERL, and (c) the reward map and value map of VIN.

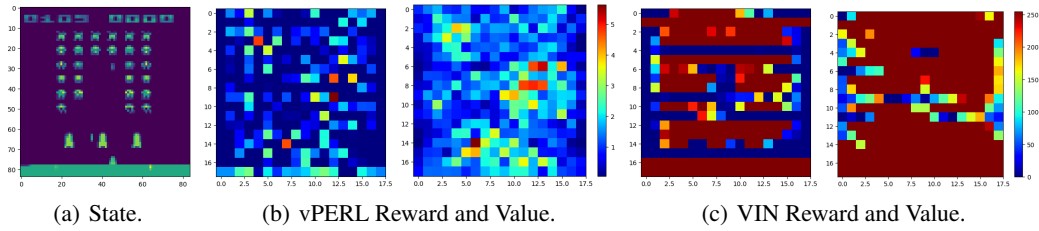

(a) State.  (b) vPERL Reward and Value.  (c) VIN Reward and Value.

Figure 22: Visualization of state, reward map and value map on Space Invaders game. (a) The state, (b) the reward map and value map of vPERL, and (c) the reward map and value map of VIN.

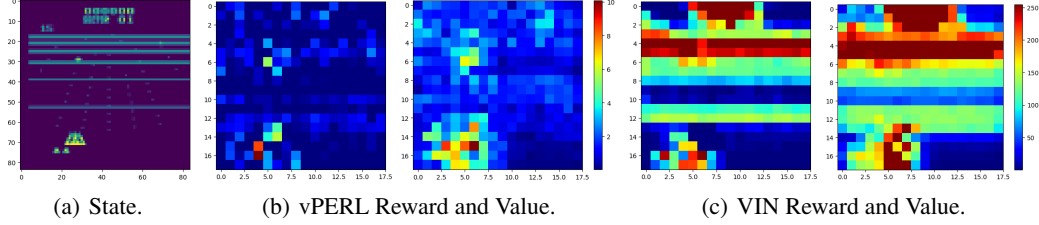

(a) State.  (b) vPERL Reward and Value.  (c) VIN Reward and Value.

Figure 23: Visualization of state, reward map and value map on Beam Rider game. (a) The state, (b) the reward map and value map of vPERL, and (c) the reward map and value map of VIN.

## F    ADDITIONAL DETAILS OF EXPERIMENTAL SETUPS

### F.1    EXPERIMENTAL SETUP OF CONTINUOUS CONTROL TASKS

Our first step was also to train a reward learning module for each continuous control task on one demonstration. To build our reward learning module for continuous tasks, we used a simple VIN and inverse VIN as the model bases of action back-tracing and transition modeling submodules, respectively. In the simple VIN model, we used 1D convolutional layer with a kernel size of 2 and

stride of 1 to implement the function $f_R$, reward map $\overline{R}$ and $Q$ value $\overline{Q}$. To accomplish the action back-tracing, the final value map of VIN was fully connected with a hidden layer with a size of 32. Reversely, we used 1D deconvolutional layer to implement the inverse VIN model. We kept the size of feature maps in both VIN and inverse VIN unchanged across all the layers. We set $K = 10$ for both the VIN and inverse VIN in all tasks. The dimension of latent variable $z$ is set to the action dimension for each task. Additionally, we used a two-layer feed forward neural network with tanh activation function as policy architecture. The number of hidden unit is set to 100 for all tasks. To extend our method on continuous control tasks, we made minor modification on the training objective. In Atari games, we used the KL divergence to measure the distance between the expert policy distribution and the action distribution in Eq. (1). In continuous control tasks, we instead directly treated the latent variable $z$ as the back-traced action and used mean squared error to measure the distance between the back-traced action and the true action in the demonstration. We set the scaling weight $\alpha$ in Eq. (1) to 1.0 for all tasks. Training was conducted with the Adam optimizer (Kingma & Ba, 2015) at a learning rate of 3e-5 and a mini-batch size of 32 for $50,000$ epochs. In each training epoch, we sampled a mini-batch of data every 20 states.

To evaluate the quality of our learned reward, we used the trained reward learning module to produce rewards, and trained a policy to maximize the inferred reward function via PPO. We trained the PPO on the learned reward function for 5 million simulation steps to obtain our final policy. The PPO is trained with a learning rate of 3e-4, a clipping threshold of 0.1, a entropy coefficient of 0.0, a value function coefficient of 0.5, and a GAE parameter of 0.95 (Schulman et al., 2016).

For a fair comparison, we used the same VIN as the model base for all the baselines. The reward function of GAIL and VAIL was chosen according to the original papers (Ho & Ermon, 2016; Peng et al., 2019). The information constraint $I_c$ in VAIL was set to 0.5 for all tasks. To enable fast training, we trained all the imitation methods with 16 parallel processes.

## F.2 Additional details of GAIL and VAIL

The *discriminator* for both GAIL and VAIL takes in a state (a stack of four frames) and an action (represented as a 2d one-hot vector with a shape of ($|\mathcal{A}| \times 84 \times 84$), where $|\mathcal{A}|$ is the number of valid discrete actions in each environment) (Brown et al., 2019b). The *discriminator* outputs a binary classification value, and $-\log(D(s, a))$ is the reward. VAIL was implemented according to the repository of Karnewar (2018). The *discriminator* network architecture has an additional convolutional layer (with a kernel size of 4) as the final convolutional layer to encode the latent variable in VAIL. We used the default setting of 0.2 for the information constraint (Karnewar, 2018). PPO with the same hyper-parameters was used to optimize the policy network for all the methods. For both GAIL and VAIL, we trained the *discriminator* using the Adam optimizer with a learning rate of 0.001. The *discriminator* was updated at each policy step.

