# OpenReview forum: "Learning Efficient Planning-based Rewards for Imitation Learning"
_ICLR.cc/2021/Conference — Reject_

### Official Review · AnonReviewer1 · 2020-10-21
**Interesting approach backed up by good empirical results**

**Rating:** 6
**Confidence:** 3

**Review:**

This paper addresses the problem of imitation learning. The authors
propose a system where you first learn a surrogate reward function
from the demonstrations, and then you do RL to optimize that reward.
The surrogate reward function is the result of learned forward and
inverse dynamics models that are "planning-embedded" in the same
sense as value iteration networks.

I think this paper is quite interesting overall, and the idea seems
worth pursuing. I know that the general approach to imitation learning
of first learning a cost function from the demonstrations and then
applying RL to optimize it has a very rich history, but I have not
seen prior work that learns the cost function in the manner proposed
in this paper. Additionally, the experimental results seem good to me,
especially on the chosen Atari games. I think the result that vPERL
achieves performance far greater than that of the demonstrations, in
some instances, will be of interest to the community. Therefore, I am
leaning toward recommending acceptance, but I am not familiar enough
with the scope of related literature to be certain.

I do have some questions which I hope the authors can address in the rebuttal.

1. What is special about Kung-Fu Master and Breakout that causes vPERL
to not perform well? Have the authors tried Atari games other than the
8 shown? Why were these 8 picked?

2. Regarding Figure 1, I'm a bit unclear about the interpretation. A
value function is a function mapping states to values; what does it
mean to have a "value map" for a single state? Is the idea that these
are spatial domains where an agent is moving around, and so the map
you're showing us implicitly represents the value of a state where the
agent is at that location?

3. After Equation 1, you say that "\pi_E(a_t | s_t) is the expert
policy distribution." But where do we get this distribution from? We
only have a dataset of expert behavior, and states are continuous, so
we will never see the same state twice in the data. So how can we
build a distribution over what actions the expert would take from any
given state? Will this distribution just be a Dirac in practice?

4. I am curious how important the use of both forward and inverse
models is in the reward learning module. The authors provide some
justification why both are needed, but it would be good to support
this with an ablation experiment.

5. How sensitive is the approach to the optimality of the
demonstrations? That is, in your experiments, you generate the
demonstrations from a policy trained on 10 million PPO steps; what if
this number of steps were lower?

6. To me, an obvious baseline would be to run PPO with the exact
configuration that you use in your experiments (with 50 million
simulation steps), but using the ground truth reward instead of the
learned reward. I am curious if the authors have considered this
baseline, and how it performs.

---

> ### Author Response · Authors · 2020-11-25
> **Responses to R1**
>
> We thank the reviewer for the constructive comments! We have revised the paper according to the suggestions and would like to clarify several things to address the reviewer’s concerns.
>
> Q1: What is special about Kung-Fu Master and Breakout that causes vPERL to not perform well? Have the authors tried Atari games other than the 8 shown? Why were these 8 picked?
>
> A1: In Kung-Fu Master and Breakout, our method also achieves performance that is much better than the demonstration and close to the expert-level performance. The potential reason for why our method does not outperform GIRIL in these two games is the negative effects of the misleading game scores and number of lives in the demonstration.
>
> We randomly chose 8 widely used Atari games. In Appendix C.1, we have added two additional Atari games, Krull and Time Pilot. In both additional games, our method consistently outperforms the expert and other baselines.
>
> Q2: Regarding Figure 1, I'm a bit unclear about the interpretation. A value function is a function mapping states to values; what does it mean to have a "value map" for a single state? Is the idea that these are spatial domains where an agent is moving around, and so the map you're showing us implicitly represents the value of a state where the agent is at that location?
>
> A2: The ‘map’ has two meanings: 1) The name of ‘reward map’ and ‘value map’ is related to the feature map of a deep neural network; 2) Generally, the value map highlights the resulting region of an agent executing a specific action. In some specific game such as Q*bert, as shown in the middle Figure 5 (c) and Figure 13 (b) of Appendix D, the highlighted region also represents the location of the agent in the state.
>
> Q3: After Equation 1, you say that "\pi_E(a_t | s_t) is the expert policy distribution." But where do we get this distribution from? We only have a dataset of expert behavior, and states are continuous, so we will never see the same state twice in the data. So how can we build a distribution over what actions the expert would take from any given state? Will this distribution just be a Dirac in practice?
>
> A3: In practice, we employ the Dirac distribution of the demonstration data, which results in a cross entropy loss for discrete action. The one hot vector of action serves as a label similar to classification tasks.
>
> Q4: I am curious how important the use of both forward and inverse models is in the reward learning module. The authors provide some justification why both are needed, but it would be good to support this with an ablation experiment.
>
> A4: We have added an ablation in Table 3 to study the contribution of each component of our method. In Table 3, PERL uses both models but is trained without the variational objective in Eq. (1). PERL can outperform the demonstration in the Centipede game, which indicates the potential importance of using both models for learning rewards. More importantly, by using the variational objective, we are able to train the planning-based action back-tracing and transition modeling in a unified generative solution. The use of ELBO in Eq. (1) greatly improves the generalization ability of our reward model.
>
> Q5: How sensitive is the approach to the optimality of the demonstrations? That is, in your experiments, you generate the demonstrations from a policy trained on 10 million PPO steps; what if this number of steps were lower?
>
> A5: Our method is not very sensitive to the optimality of the demonstrations. We have added an ablation in Appendix B.2 to study the effect of the optimality. Table 9 and Table 10 show the average return of vPERL-Small-R_Mean with demonstrations of different optimality on Atari games and continuous control tasks, respectively. In the experiments, we trained a PPO agent with the ground-truth reward for 10 million (10M) simulation steps as the expert for Atari games, and 1 million (1M) steps for continuous control tasks.
>
> To study the effects of optimality of the demonstrations, we additionally trained PPO agents with fewer simulation steps: 1M steps and 5M steps for Atari games, and 0.1M steps and 0.5M steps for continuous control tasks. With the additional suboptimal PPO agents, we generated demonstrations with 10%, 50% optimality of the 10M-step ‘Expert’ for both Atari games and continuous control tasks. The results show that our method outperforms the expert by a large margin in Atari games and reaches the demonstration-level performance in continuous control tasks with demonstrations of different optimality.

---

> > ### Author Response · Authors · 2020-11-25
> > **Responses to R1 (continued)**
> >
> > Q6: To me, an obvious baseline would be to run PPO with the exact configuration that you use in your experiments (with 50 million simulation steps), but using the ground truth reward instead of the learned reward. I am curious if the authors have considered this baseline, and how it performs.
> >
> > A6: We have added the results of PPO training with the true reward for 50 million simulation steps in Table 2.  Given more sufficient training steps than the expert, our reward function can generalize to the regions not covered by the demonstration to some extent. Therefore, our method can still achieve a performance that is better than it in Battle Zone, Centipede, and Q*bert, and close to it in Seaquest and Beam Rider. However, the performance of other baselines is much worse than the oracle.

---

### Official Review · AnonReviewer2 · 2020-10-23

**Rating:** 6
**Confidence:** 2

**Review:**

This paper presents a inverse RL method that utilizes a differential planning module and dynamics modeling to learn reward signals. The learned reward function can be used to train policies that can match or even exceed demonstration, as demonstrated in several Atari games and two simple continuous control tasks.

The paper demonstrates a novel usage of VIN like planning module for inverse RL and the result is quite compelling. Although the writing can and should be improved to help reader to understand what is going on, including multiple grammatical errors like " we introduce a variational solution to optimize the both submodules jointly".

Some technical questions:
(1)$p_\theta$ takes $z$ and $s_t$ as input, but in equation (1), it also only takes $s_t$ as input. I am a bit confused here.
(2) As far as I understand, the learned transition dynamics is only meaningful for the expert action, for example, given $s_t$, if the $a_t$ is completely different from the expert action, it is likely to predict completely wrong $s_{t+1}$, and thus completely wrong reward estimate with the setup in the paper. Since the policy is able to learn much better with the learned reward function, I will imagine it learns to take action that is completely different from the demonstration. Any intuitive explanation why this will work so well? It will be also interesting to see how the learned reward compare to the true reward in those region not covered by the demonstration.
(3) The continuous control tasks is like a toy example, but I guess the impressive part is that it can learn so well with only 1000 demonstration data. But are there any challenges to scale this to more complicated tasks given enough demonstration? If not, potential limitation should be discussed.

---

> ### Author Response · Authors · 2020-11-25
> **Responses to R2**
>
> We thank the reviewer for the constructive comments! We have revised the paper according to the suggestions and would like to clarify several things to address the reviewer’s concerns.
>
> Q1: pθ takes z and st as input, but in equation (1), it also only takes st as input. I am a bit confused here.
>
> A1: Our variational objective in equation (1) follows the formulation of conditional variational autoencoder (VAE) in [1]. The basic formulation of a conditional VAE includes a conditional encoder, decoder and prior. In our formulation, we regard the action back-tracing module $q_\phi(z|s_t, s_{t+1})$ as an encoder, transition modeling $p_\theta(s_{t+1}|z, s_t)$ as a decoder, and $p_\theta(z|s_t)$ as the prior. We have clarified this in Section 3.1.
>
> [1] Kihyuk Sohn, Honglak Lee, Xinchen Yan. Learning Structured Output Representation using Deep Conditional Generative Models. In NIPS 2015.
>
> Q2: As far as I understand, the learned transition dynamics is only meaningful for the expert action, for example, given $s_t$, if the $a_t$ is completely different from the expert action, it is likely to predict completely wrong $s_{t+1}$, and thus completely wrong reward estimate with the setup in the paper. Since the policy is able to learn much better with the learned reward function, I will imagine it learns to take action that is completely different from the demonstration. Any intuitive explanation why this will work so well? It will be also interesting to see how the learned reward compares to the true reward in those regions not covered by the demonstration.
>
> A2: When a state is completely different from the states in the demonstration, it is a ground challenge for all imitation learning methods because the accessible information is extremely limited. For example, given the one-life demonstration of the Seaquest game, the demonstration only contains the states and actions with the submarine floating on the water. It is hard for a reward function to guide the agent to take action to dive and hit to achieve a higher score. Table 2 shows that even in such a challenging task, our method still outperforms than other baselines because of  its better generalization ability.  Other methods suffer from the overfitting issue.
>
> The generative intrinsic reward driven imitation learning (GIRIL) uses the prediction error of state as a surrogate reward, which highly relies on the transition dynamics. When the state and action are completely different from the demonstration, the prediction error based surrogate reward would be ambiguous. Since it encourages the agent to visit a state that is away from the state prediction, which can lead to a policy either better or worse than the demonstrator.
> In contrast, our method learns a straightforward surrogate reward via learning to plan from the demonstration data. We achieve this goal by optimizing a planning module and transition modeling module in a unified generative solution, which is based on the objective of conditional VAE. The ELBO of conditional VAE greatly improves the generalization ability and smoothness of the surrogate reward function, which enables our reward function generalizes well locally from demonstration. This is critical for our method to consistently perform well in diverse Atari games.
>
> As illustrated in Figure 2, during policy optimization, we calculate the rewards independent of action, and the predicted $s_{t+1}$. Given a state that is not very far away from the one in the demonstration, our planning-based reward function is smooth enough that it can output a good reward signal to guide the agent to  execute a reasonable action.
>
>
>
>
> In the 4th column of Table 2, we have added the results of the PPO agent trained with ground-truth reward for 50 million steps, i.e. PPO-50M. PPO-50M includes the region not covered by the demonstration, and therefore it serves as an oracle of imitation learning performance. Given more sufficient training steps than the expert, our reward function can generalize to the regions not covered by the demonstration to some extent. Therefore, our method can still achieve a performance that is better than the oracle in Battle Zone, Centipede and Q*bert, and close to it in Seaquest and Beam Rider. However, the performance of other baselines is much worse than the oracle.
>
> Q3: The continuous control tasks is like a toy example, but I guess the impressive part is that it can learn so well with only 1000 demonstration data. But are there any challenges to scale this to more complicated tasks given enough demonstration? If not, potential limitation should be discussed.
>
> A3: Thanks for the valuable suggestion. Although the results in section 4.3 and Appendix B.1 show that our method is not sensitive to numbers of demonstrations, the main focus of this paper is imitation learning with limited demonstration data. We remain the extension of our method to more complicated continuous control tasks as future work in the conclusion.

---

### Official Review · AnonReviewer3 · 2020-10-24
**Interesting approach, needs more empirical analysis**

**Rating:** 6
**Confidence:** 3

**Review:**

This paper proposes a method for inverse reinforcement learning that incorporates a differential planning module. Explicit transition dynamics modeling with inverse value iteration is added to promote meaningful reward learning. Empirical evaluations on several high-dimensional Atari environments and 2 continuous control environments are provided which show improvements over existing inverse reinforcement learning baselines when given only one-life demonstrations. Some visuals are also presented to show that the proposed method is able to learn more meaningful reward maps than previous methods.

Positives:
1. The proposed method is natural and makes use of a few existing works. The combination of Value Iteration Networks (VIN) and the Generative Intrinsic Reward driven Imitation Learning (GIRIL) is novel.

2. The visual reward maps show that the proposed method learns a more semantically meaningful reward map than VIN. When using PPO on the learned reward function, it learns a better performing policy faster in most cases.

3. The results in continuous control environments demonstrate generalizability.

Negatives:
1. There is little analysis of the learned reward maps beyond a few plots for visual comparisons. As the proposed method bears close connections to VIN (used as a submodule) and GIRIL (same joint learning pipeline), more detailed comparisons are definitely needed. A few concrete suggestions are:
a) add reward maps for GIRIL and provide qualitative analysis on the difference, e.g., in semantic meanings.
b) for the proposed method, there are two variants on using the predicted reward maps: mean and max. It is unclear which version of the maps are shown in the paper and the empirical results are not conclusive on which one is better. It would be interesting to analyze the differences between them and, as a bonus, provides a rule-of-thumb on which one to use.

2. There are several places where the writings could be improved. Some missing derivation details make understanding the paper difficult.

Questions:
1. Please clarify a few terms in the paper:
a) Embedded MDP in section 3.1 near the bottom of page 3
b) What is the conditional VAE? And the derivation details of equation (1) in the paragraph about Variational solution to vPERL.

2. It is somewhat counterintuitive to me that vPERL-small generally outperforms vPERL-large since the small version has less state information. Could the authors provide some explanation for this?

3. The baseline method VIN/BC is confusing. Could the authors clarify whether it means Behavioral cloning (as in supervised learning with state-action pairs from demonstrations) or VIN?

======================
Post-rebuttal comments:

I want to thank the authors for providing clear answers to my questions and comments. I found the answers satisfactory so I raised my score to 6.

---

> ### Author Response · Authors · 2020-11-25
> **Responses to R3**
>
> We thank the reviewer for the constructive comments! We have revised the paper according to the suggestions and would like to clarify several things to address the reviewer’s concerns.
>
> Q1: Please clarify a few terms in the paper: a) Embedded MDP in section 3.1 near the bottom of page 3 b) What is the conditional VAE? And the derivation details of equation (1) in the paragraph about Variational solution to vPERL.
>
> A1: a) VIN assumes there is some unknown MDP $\overline{M}$ where the optimal plan in $\overline{M}$ contains useful information about the optimal plan in the original MDP $M$.  To distinguish from the original MDP, the unknown MDP $\overline{M}$ is referred to as embedded MDP. This has been included in the details of VIN in Section 2.
>
> [1] Aviv Tamar, Yi Wu, Garrett Thomas, Sergey Levine, Pieter Abbeel, Value Iteration Networks. In NIPS 2016.
>
> b) We have added more details of equation (1) in Section 3.1. A  variational  autoencoder  (VAE) can be defined as being an autoencoder whose training is regularised to avoid overfitting and ensure that the latent space has good properties that enable generative process. To avoid the learned planning-based reward overfit to the demonstration, we optimize both submodules in a unified variational solution, which follows the formulation of conditional VAE , which is a conditional generative model for structured output prediction using Gaussian latent variables. Conditional VAE is built up with a conditional encoder,decoder and prior.  Accordingly, we regard the action back-tracing module $q_\phi(z|s_t, s_{t+1})$ as the encoder, $p_\theta(s_{t+1}|z, s_t)$ as the decoder, and $p_\theta(z|s_t)$ as the prior.
>
> The first two terms on the RHS of Eq. (1) in the first line denote the evidence lower bound (ELBO) of the conditional VAE (Sohn et al., 2015). These two terms are critical for our reward learning module to perform planning-based action back-tracing and transition modeling. Additionally, we integrate the third term on the RHS of Eq. (1) in the second line to further boost the action back-tracing. The third term minimizes the KL divergence between the expert policy distribution πE(at|st) and the action distribution qφ(ˆat|st,st+1), where ˆat=Softmax(z) is transformed from the latent variable $z$. In this way, we train the forward state transition and action back-tracing simultaneously.
>
> Q2: Why vPERL-Small outperforms vPERL-Large?
>
> A2: vPERL-Small has fewer trainable parameters than vPERL-Large. With smaller complexity, vPERL-Small generalizes better than  vPERL-Large when learning from the limited demonstration data. Therefore, vPERL-Small performs better than vPERL-Large.
>
> Q3: Clarify the baseline method VIN/BC.
>
> A3: In the original paper [1], Tamar et al. applied VIN for imitation learning in terms of supervised learning, which is similar to behavior cloning, For better clarity, we have changed  ‘VIN/BC’ to ‘VIN’.
>
> [1] Aviv Tamar, Yi Wu, Garrett Thomas, Sergey Levine, Pieter Abbeel, Value Iteration Networks. In NIPS 2016.
>
> Q4: Add reward maps for GIRIL and provide qualitative analysis on the difference, e.g., in semantic meanings.
>
> A4: We have added reward maps for GIRIL in Figure 5. Figure 5 shows the state, the reward maps of vPERLand GIRIL in three Atari games. The reward map of GIRIL can be close to zero (in Battle Zone) andstate (in Q*bert) or occasionally informative (in Centipede), which is ambiguous and less informative. In contrast,  the reward map of our vPERL is more straightforward,  and consistently attends to informative regions in the state for all of the games. The informative regions indicate the resulting region of the agent executing an action. This means that our method learns a meaningful reward function along with useful planning computation.
>
> Q5: For the proposed method, there are two variants on using the predicted reward maps: mean and max. It is unclear which version of the maps are shown in the paper and the empirical results are not conclusive on which one is better. It would be interesting to analyze the differences between them and, as a bonus, provides a rule-of-thumb on which one to use.
>
> A5: Generally, vPERL-Small performs better than vPERL-Large, and the training of vPERL-Small is faster since it has fewer trainable parameters. Though vPERL-Large also outperforms other baselines, we can use vPERL-Small by default for better efficiency.

---

### Official Review · AnonReviewer4 · 2020-10-29
**Learning a policy from one demonstration without rewards**

**Rating:** 5
**Confidence:** 3

**Review:**

This paper assumes no access to the reward values and attempts to learn a policy by starting just with one demonstration to define the reward. For obtaining the reward, the authors rely on the ideas from Value Iteration Networks (VIN) method and they add the modules that help to deal with cases with complex transition dynamics. The resulting method is tested on atari domain and on continuous control tasks.

Strong points:

- The ideas from Value Iteration Networks are applied to the environments with complex dynamics.
- The problem setting is interesting. The restrictive assumption about the amount of demonstrations makes the method practical.
- The results are promising and seem to significantly outperform the competing baselines from inverse reinforcement learning.

Weak points:

- The paper relies a lot on VIN method ideas without introducing it with sufficient detail. This makes it difficult to follow as many  concepts (and notation) in Section 3 are not properly introduced. Besides, the contributions of this paper are not very clearly stated.
- The analysis part in 4.1 is not very informative and does not provide *experimental* evidence on *why* the proposed method works well and what issues with previous methods it addresses. For example, it is still not clear to me why the small model performs better than the big model in Table 2. Another example is the lack of ablations to understand the contribution of each of the components of the method. Additionally, the reader would benefit from understanding how the gap between the proposed method and the baselines changes with the growing amount of demonstrations: In particular, does the proposed model address the issues of low data regime or does it help in general?
- The atari environment actually includes the scores in the input images (clearly in Breakout, Bean Rider, Space Invader, Seaquest, Kung Fu, and maybe in Qbert, BattleZone). Is it fair if no reward is assumed to be available?
- The paper is quite hard to understand, it immediately jumps into details of the method in Section 3 without providing sufficient overview and intuition into what the method tries to achieve and what the issues with previous works are.


I am leaning towards the rejection of this paper. While the experimental results are very encouraging as they are provided for the environments with complex dynamics and with very little expert data, my main concerns are that 1) the method description is currently very hard to understand, the contributions are not stated clearly, and 2) the experimental results do not include any informative analysis to understand the underlying reasons for why the method performs well.

Questions:
- In the conclusion, the authors say "policies that generalise well to new tasks", what is meant by this and what are the supporting experiments?
- Why does the small method model work better than the big model?
- What is the contribution of different components of the method? How sensitive the method is to the choice of hyperparameters, such as K?
- In Figure 4 it seems that many methods didn't converge to the final score yet. Why was such a number of training steps chosen? What happens if the experiments run for longer?
- What is the computational complexity of the proposed algorithm?
- Why is it possible to take average or max reward values of the reward map? How is the reward map idea justified when it is provided not for the map of states but where each map corresponds to one state?

Additional comments
- The clarity of writing could be further improved. For example, see the Conclusion section (but it applies in other parts of the paper as well), avoid -> avoids, outperform -> outperforms, limiting and limited are repeated etc. Figure 3 has its labels shifted and Figure 1 seems to have the rows swapped. In Figure 2: is q_\phi meant?
- Maybe the analysis part in 4.1 is more suitable for the related work.

=== Post rebuttal ===

I would like to thank the authors for the very detailed response and the improvements in the manuscript. I found the ablation experiments and experiment with "No scores" particularly useful. However, I am still a bit confused about better performance of the smaller model. Maybe, more understanding could be gained if there was an additional "tiny" model that would show that when going beyond a certain size, the performance degrade. Additionally, if the overfitting is an important issue, some regularisation methods could be explored.

Finally, in the light of many changes in the paper and the original request of all the reviewers to have the writing in the manuscript improved, I think the paper would benefit from another round of reviews. However, I find this method promising and if the paper is not accepted this time, I encourage the authors to re-submit a revised version.

---

> ### Author Response · Authors · 2020-11-25
> **Responses to R4**
>
> We thank the reviewer for the constructive comments! We have revised the paper according to the suggestions and would like to clarify several things to address the reviewer’s concerns.
>
> Q1: Provide sufficient overview and intuition into what the method tries to achieve and what the issues with previous works are.
>
> A1: In this paper, we focus on addressing the two key issues of previous methods when learning with limited demonstration data,  i.e.,  1) overfitting problem,  and 2) ambiguous quality of the reward function. To address these issues, we propose to learn a straightforward surrogate reward function by learning to plan from the demonstration data, which is more reasonable than the previous intrinsic reward function (i.e., the prediction error between states).
>
> 1) Most state-of-the-art IRL methods fail to meet the performance of demonstrations in high-dimensional environments with limited demonstration data, e.g., a one-life demonstration in Atari domain. This is due to the main goal of these IRL approaches is to recover a reward function that justifies the demonstrations only. The rewards recovered from limited demonstration data would be vulnerable to the overfitting problem. Optimizing these rewards from an arbitrary initial policy results in inferior performance.
>
> 2) Recently, Yu et al. (2020) proposed generative intrinsic reward driven imitation learning (GIRIL) method for learning with limited demonstration data. This method outperforms expert and IRL methods in Atari games. Although GIRIL uses the prediction error as curiosity to design the surrogate reward that encourages (pushes) states away from the demonstration and avoids overfitting, the curiosity also results in ambiguous quality of the rewards in the environment. Here, the ambiguous rewards can lead to a policy either better or worse than the demonstrator.  For example, Table 2 shows that GIRIL performs worse than demonstration in the Space Invaders game. A random state can also be assigned with a large reward, as a result, the prediction error based surrogate reward may lead to a random policy which may be much worse than the demonstration.
>
> We employ the reward function in the embedded VIN module as a straightforward surrogate reward, which does not have the issue that always pushes the state away from the demonstration. Instead, our surrogate reward function is learned along with a planning computation learning in an embedded differential planning module.  Thus, our method  is more reliable than the prediction error-based surrogate reward, and has consistent semantic meaning with the ground-truth reward in the MDP.
>
> Q2: Provide sufficient details of VIN, and clearly state the contribution.
>
> A2: Thanks for the valuable suggestions. We have added more details of VIN at the end of Section 2.
>
> Value iteration (VI)  is a well-known method for calculating the optimal value $V^*$ and optimal policy $\pi^*$: $ V_{n+1}(s) = \max_a Q_n(s, a)$, where $Q_n(s,a) = R(s,a) + \gamma\sum_{s'}T(s'|s,a)V_n(s')$ denotes the Q value in the $n$th iteration.
> The value function $V_n$ in VI converges as $n\rightarrow\infty$ to $V^*$, from which the optimal policy may be derived as $\pi^*(s)=\arg\max_a Q_\infty(s,a)$.
>
> Value iteration networks (VIN) proposes to embed value iteration (VI) process with a recurrent convolutional network, and generalizes well in conventional navigation domains. VIN assumes there is some unknown embedded MDP $\overline{M}$ where the optimal plan in $\overline{M}$ contains useful information about the optimal plan in the original MDP $M$. VIN connects the two MDPs with a parametric reward function $\overline{R}=f_R(s)$.
>
> Our main contribution is that we propose a novel variational planning-embedded reward learning method to address the two issues  of previous methods.
>
> 1) To tackle the ambiguous reward issue, we propose to learn a straightforward surrogate reward function by learning to plan from the demonstration data, which is more reasonable than the previous intrinsic reward function (i.e., the prediction error between states). We employ the reward function in the embedded VIN module as a straightforward surrogate reward, which does not have the issues that always push the state away from the demonstration.  Thus, our method  is more reasonable than the prediction error-based surrogate reward. Moreover, our surrogate reward has consistent meaning with the ground-truth reward in a MDP.
> 2) For the overfitting issue, we leverage a variational objective based on the conditional variational autoencoder (VAE) to jointly optimize the planning module and transition modeling, which greatly improves the generalization ability. This is critical for the success of achieving straightforward and smooth reward function with limited demonstration data.

---

> > ### Author Response · Authors · 2020-11-25
> > **Responses to R4 (continued1)**
> >
> > Q3: Provide informative analysis and experimental evidence on why the proposed method works well and what issues with previous methods it addresses. Add ablations to study the contribution of different components of the method.
> >
> > A3: We have provided informative analysis and experimental evidence in Section 4.1.1.
> > In Section 4.1.1, we study the contribution of each component of our method,  i.e.   Action Back-tracing submodule,  Transition Modeling Submodule, and the variational objective in Eq. (1). Specifically, we directly train the Action Back-tracing and Transition Modeling submodules in terms of supervised learning. We used the mean of the reward map and prediction error of the next state as the reward for the former and latter submodules, respectively. To study the contribution of the variational objective, we introduced another baseline, PERL, which trains both submodules as an autoencoder.
> >
> > Table 3 shows that individual training of each component obtains impaired results; to no avail. PERL successfully outperforms the demonstration in one game, i.e. Centipede, which indicates the potential advantage of using both submodules. However, PERL totally fails in the other seven games, while vPERL outperforms the demonstration in eight games and outperforms expert in six. The large performance gap between PERL and vPERL indicates the variational objective in Eq. (1) is important to learn efficient rewards. To further investigate the key reason for why our method works well, we added another baseline - supervised PERL, which forces the encoding of PERL to be close to the true action. The supervised PERL fails in all of the games.  Comparing with vPERL, we can attribute the critical contribution to the use of the ELBO of conditional VAE, or more specific, the term ${\rm KL}(q_\phi(z|s_t, s_{t+1})\|p_\theta(z|s_t))$ in Eq. (1). It helps vPERL to work well and outperform previous methods for two reasons:
> > 1)  The generative training of VAE can serve as a good regularization to alleviate the overfitting problem.
> > 2) The regularization term enables vPERL to learn a smooth value function and reward function, which consistently provides straightforward and informative rewards for the moving states in the environment.
> >
> > Experimental evidence:
> > 1) Better generalization ability. The empirical results in Table 2 and Figure 4 show that VIN, GAIL and VAIL are vulnerable to overfitting problem, usually achieves poor results, to no avail, and has fewer chances to reach the demonstration-level performance. In contrast, our vPERL has better generalization ability and consistently achieves performance that is either close to or better than the expert.
> > 2) Straightforward and informative reward. Figure 5 shows the state, the reward maps of vPERL and GIRIL in three Atari games. The reward map of GIRIL can be close to zero (in Battle Zone) and state(in Q*bert) or occasionally informative (in Centipede), which is ambiguous and less informative. In contrast, the reward map of our vPERL is more straightforward, and can consistently attend to informative regions in the state for all of the games.
> > The informative analysis and experimental evidence indicate that our method addresses two issues of previous methods: 1) overfitting problem,  and 2) ambiguous quality of the reward function. Therefore, our method can outperform previous methods.
> >
> > Q4: Clarify “policies that generalize well to new tasks”
> >
> > A4: Thanks for your suggestion. This means our method can learn a reward function with better generalization ability.
> >
> > Q5: Why does vPERL-Small perform better than vPERL-Large?
> >
> > A5: vPERL-Small has fewer trainable parameters than vPERL-Large. With smaller complexity, vPERL-Small generalizes better than  vPERL-Large when learning from the limited demonstration data. Therefore, vPERL-Small performs better than vPERL-Large.
> >
> > Q6: How sensitive the method is to the choice of hyperparameters, such as K?
> >
> > A6: We have added an ablation study on K in Appendix B.3. To study the effects of the hyperparameter K, we evaluate our method on two Atari games and two continuous control tasks with two additional K (K=5, and K=15). Table 11 and Table 12 shows the average return of vPERL-Small-R_Mean versus different numbers of K on Atari games and continuous control tasks. With the three choices of K, our method consistently outperforms the expert in the Atari games, and reaches the demonstration-level (best) performance in continuous control tasks. This indicates that our method is not very sensitive to the choices of hyperparameter K.

---

> > > ### Author Response · Authors · 2020-11-25
> > > **Responses to R4 (continued 2)**
> > >
> > > Q7: Why was such a number of training steps chosen? What happens if the experiments run for longer?
> > >
> > > A7: We chose the number of training steps following [1]. Within such a number of training steps, our method outperforms other baselines and is much more sample-efficiency in most cases in Figure 4 and Figure 6. Due to the limited time and accessible computation resources in the current stage, we are unable to run longer experiments for all of the Atari games. We  will add this in the final version.
> > >
> > > [1] Daniel S. Brown et al. Extrapolating Beyond Suboptimal Demonstrations via Inverse Reinforcement Learning from Observations, In ICML 2019.
> > >
> > > Q8: What is the computational complexity of the proposed algorithm?
> > >
> > > A8: For Atari games, it takes vPERL-Small 0.000289 seconds and vPERL-Large 0.000374 seconds to process a data sample. For continuous control tasks, it takes vPERL 0.000427 seconds to process a data sample, which is fast.
> > >
> > > Q9: Why is it possible to take average or max reward values of the reward map?
> > >
> > > A9: Since the learned reward maps of both vPERL-Large and vPERL-Small clearly attend to meaningful regions of an executed action. For each state, vPERL learns useful planning computation and the information has been reflected on the reward map. We use mean aggregation or max aggregation to summarize the information from the reword map. Therefore, we can take advantage of such useful planning computation information in the reward map to train an agent without access to the true reward.
> > >
> > > Q10: Does the proposed model address the issues of low data regime or does it help in general? Add ablation study on the growing number of demonstrations.
> > >
> > > A10: Following the valuable suggestion, we have added an ablation study on the growing number of demonstrations in Appendix B.1. Table 5 and Table 6 show the detailed quantitative comparison of imitation learning methods across different numbers of full-episode demonstrations in the Centipede game and Qbert game. The comparisons on two continuous control tasks, InvertedPendulum and InvertedDoublePendulum, have been shown in Table 7 and Table 8. The results show that our method vPERL achieves the highest performance across different numbers of full-episode demonstrations.
> > >
> > > Q11: The Atari environment actually includes the scores in the input images. Is it fair if no reward is assumed to be available?
> > >
> > > A11: In imitation learning, we assume the agent has no explicit access to the ground-truth reward signals. To avoid the potential effects of the game scores and number of left lives in the states of demonstrations, we have added ablation study in Table 14 of Appendix C.2. Following [1], we generate the “No-score Demo.” by masking out the game score and number of left lives in the demonstrations. Table 14 compares the performance of our method (vPERL-Small-R_Mean) on both “Standard Demonstration” and “No-score Demo.”.
> > >
> > > The results show that our method achieves better performance on the “No-score Demo.” than the “Standard Demo.”. Our method can learn to outperform the expert without explicit access to the true rewards, and does not rely on the game scores and the number of left lives in the states of demonstrations. Furthermore, the results show that the performance of our method can be improved by masking out the game scores and the number of left lives in the demonstrations.
> > >
> > > Q12:  Figure 3 has its labels shifted and Figure 1 seems to have the rows swapped. In Figure 2: is q_\phi meant? Maybe the analysis part in 4.1 is more suitable for the related work.
> > >
> > > A12: Thanks. We have fixed the figures and revised section 4.1 and the related work section.

---

### Decision · Program_Chairs · 2021-01-07
**Final Decision**

**Decision:**

Reject

**Comment:**

The authors introduce vPERL, a model that generates an intrinsic reward for imitation learning. vPERL is trained on demonstrations to minimise a variational objective that matches a posterior formed by "action backtracking" and a forward model, with the intrinsic reward coming from the reward map. The authors might be interested in related work on few shot imitation learning: e.g., "One shot imitation learning", Duan et al, 2017, "Watch, try learn: meta-learning from demonstrations and rewards", Zhou et al 2019. As all reviewers pointed out, and I can confirm, the paper is quite tricky to understand in its present form, and would very much benefit the writing being re-visited to more clearly express the ideas within (in particular, section 3, which is the core of the contributions).